# Linker histones consolidate heterogenous nucleosome fiber contacts by linking together multiple nucleosomes

Zenita Adhireksan[1], Deepti Sharma[1], Qiuye Bao[1], Phoi Leng Lee[1], Sivaraman Padavattan [1,5], Gabriela E. Davey[1], Jeffrey C. Hansen[2] & Curtis A. Davey [1,3,4] ✉

The consensus mode for linker histone (H1) association coincides with 'on-dyad' binding to an individual nucleosome, making it challenging to rationalize the chromatin dynamics and compacting activities of H1 in the context of a highly heterogeneous structural scaffold. Here, we investigate the activity of the somatic H1 variants by conducting crystallographic analysis of nucleosomal assemblies and characterization of nucleosome array condensates, which recapitulate long-range nucleosome fiber interactions in chromatin. H1 is observed to associate variant-dependently with nucleosomes through a diversity of binding modes that include linking multiple nucleosomes/fibers together. Binding versatility is facilitated by the proclivity of the H1 globular domain to recognize DNA structural motifs, which are similar between an individual nucleosome and specific niches within clusters of nucleosomes. We propose that linker histones support a structurally and functionally complex repertoire for chromatin regulation by assuming a variety of context-and variant-dependent DNA binding modes.

The eukaryotic genome consists of chromatin fibers, which can comprise more than one million nucleosomes in tandem for a given chromosome[1,2]. Condensing nucleosomes into compact states provides a means for gene silencing in heterochromatin, maintaining genomic integrity, and generating mitotic chromosomes for cell division. Nucleosome compaction is facilitated by the action of a variety of chromatin architectural factors, which include linker histone (LH), cohesin, condensin, and HP1, amongst other factors, that appear to act at different levels of the chromatin structural hierarchy. While it has come to light in recent years that compact chromatin structure in vivo generally consists of irregular interdigitated nucleosome fibers with zigzag features, as opposed to highly regular '30 nm fiber' structures, the mechanisms by which architectural factors operate to shape local chromatin 3D organization are not well understood[3].

As the most abundant chromatin-binding protein, LHs are essential factors in higher eukaryotes, with eleven H1 variants present in mammals, of which seven (H1.1–H1.5, H1.0, and H1x) are somatic, and four (H1oo, H1t, H1t2, and HILS1) are germ cell-specific[4]. LH variants display specialized genomic functions and gene-specific regulatory activities as well as cell-dependent expression and differential chromatin localization[5]. Notably, H1.0 has been shown to play a key role in controlling proliferative potential and differentiation status of tumor cells[6], while cancer-associated mutations in the H1.2–H1.5 variants can act as oncogenic drivers, rendering the LHs bona fide tumor suppressors[7].

[1]School of Biological Sciences, Nanyang Technological University, Singapore, Singapore. [2]Department of Biochemistry and Molecular Biology, College of Natural Sciences, Colorado State University, Fort Collins, CO, USA. [3]NTU Institute of Structural Biology, Nanyang Technological University, Singapore, Singapore. [4]Małopolska Centre of Biotechnology, Jagiellonian University, Krakow, Poland. [5]Present address: Department of Biophysics, National Institute of Mental Health and Neurosciences, Bangalore, India. ✉e-mail: curtis.davey@uj.edu.pl

LH structure is generally conserved across protein variants in metazoans, with a short N-terminal domain ( ~ 30 residues; $LH_{NT}$) and a long C-terminal domain (~110 residues; $LH_{CT}$) that are unstructured in solution, linked by a globular domain ($LH_{GD}$) of about 80 amino acids[4,8,9]. The $LH_{GD}$ consists of a so-called winged helix motif, having three off-parallel packed α-helices. Although the $LH_{GD}$ is essential and alone sufficient for nucleosome association, the $LH_{CT}$ is required for high-affinity chromatin binding and condensing activity[10–13]. There have been several models proposed for how the LHs bind to chromatin, which differ largely in whether the $LH_{GD}$ assumes a roughly central position on the nucleosome dyad, interacting with both linker DNA arms (on-dyad), or is situated off to one side, interacting preferentially with one of the linker DNA arms (off-dyad)[9].

In conjunction with advances in cryogenic electron microscopy (cryo-EM), there have been many structures solved of nucleosomes assembled with different LH variants and truncatants (reviewed in ref. 14), all of which have on-dyad association of the $LH_{GD}$, with two exceptions. Cryo-EM structures of glutaraldehyde-fixed 12-nucleosome arrays assembled with either human H1.4 or H5 (a.k.a. avian H1.0) display a 2-start, zig-zag 30 nm fiber configuration, in which the $LH_{GD}$ associates in an off-dyad fashion[15,16]. Although footprinting analyses of H1.5 or H1.0 binding to multi-nucleosome (3- or 6-nucleosome) arrays have instead shown on-dyad association[17,18], this off-dyad H1 binding is analogous to the binding mode elucidated in an earlier solution NMR study yielding a *Drosophila* H1$_{GD}$ (dH1$_{GD}$)–nucleosome structure[19]. However, with the exception of the 30 nm fiber, cryo-EM structures have been based on individual nucleosome-LH assemblies (chromatosomes) in isolation, and the few crystal structures reported coincide with discontinuity/interruption of the double helix between chromatosomes in the lattice as well as limited resolution.

It appears to have been largely assumed that the physiological substrate for an LH is a single nucleosome, but this may not be accurate considering the salient chromatin compacting activities of LHs and the fact that diverse domains of the core histones participate avidly in the condensation of chromatin fibers by mediating internucleosomal interactions[20]. Therefore, from the standpoint of chromatin association, it remains to be established what LH binding modes are prevalent in vivo, in the context of crowded, condensed, or compact poly-nucleosomal states.

In order to shed light on the nature of LH association with nucleosome fibers as well as the structure of compacted chromatin, we developed a strategy for designing nucleosomes to self-assemble into continuous fibers[21]. Application of this approach so far entailed crystallizing nucleosomes or dinucleosomes having 4-nucleotide overhangs, which foster Watson-Crick base pairing to yield a continuous double helix that comprises linker DNA connecting nucleosomes in the lattice. Two of the dinucleosome constructs yielded near-atomic resolution structures in which zigzag configured nucleosome fibers span the length of the crystals and are interdigitated with one another in the lattice[22] (HD animation, ref. 23). When co-crystallized with LH, in addition to on-dyad binding of H1.0, another copy of H1.0 was found associated in a newly observed 'linking' mode[22] (formerly referred to as the 'non-dyad' mode; HD animation, ref. 24.). This linking mode entails the $LH_{GD}$ associating with three separate nucleosome fibers, thereby stabilizing compaction and interdigitation. This led us to propose that LHs may foster chromatin compaction by associating with a variety of structural niches inherent to condensed nucleosome fibers.

In this study, we analyzed the LH association from the crystal structures of a variety of nucleosome-LH assemblies that stem from both our recent[21,22] as well as unpublished works. In conjunction with in vitro nucleosome and nucleosome array condensate experiments, the results suggest that condensed chromatin harbors a variety of inter-nucleosomal motifs to which LHs can associate in a variant-dependent fashion. The existence of alternative LH binding modes, which include multi-nucleosome association, is consistent with the observed promiscuity and plasticity of LHs as well as their ability to bind nucleosomes with greater than unity stoichiometry and promote interfiber crosslinking in condensates.

## Results

### Nucleosome-linker histone assembly structures

We screened 24 different cohesive-ended mononucleosome and dinucleosome constructs, of which 10 yielded well-diffracting crystals (Table 1 in ref. 21)[21,22]. Various constructs were also co-crystallized with different LH variants, yielding instances of LH-associated electron density for H1.0, H1x, H5, and H1.3 that was sufficiently clear to permit explicit modeling of the $LH_{GD}$ (Table 1; Supplementary Tables 1 and 2; Supplementary Figs. 1–4). Co-crystallization trials with H1.1, H1.2, H1.4, or H1.5 also yielded many near-atomic resolution data sets, but with insufficiently clear electron density associated with the LH proteins to permit model building. Generally, across all assemblies for which we were able to obtain data of sufficient resolution to allow a nucleosome-based molecular replacement phasing solution, LH could not be explicitly modeled (given the presence in the crystal of at least 1 bound LH molecule/nucleosome; see Methods). Nonetheless, in many data sets, residual features in the electron density maps suggest DNA-bound LHs that are associated with static and/or dynamic disorder.

With one particular type of construct, which consists of a 165 bp double helix having 4-nucleotide overhangs at both termini, we were able to solve LH-nucleosome assembly structures from crystals grown in the presence of H1x, H1.0, H5, or H1.3. The same overall lattice arrangement was obtained using three different 169 'bp' constructs (169a, 169ak, 169an) that differ by virtue of the DNA sequence associated with the single-stranded termini. The 169 nucleosomes dimerize to form pairs, in which the self-complementary, 4-nucleotide overhangs from each of the linker DNA termini are associated with Watson-Crick base pairing (Fig. 1a and b). This results in a pseudo-continuous double helix of 338 bp encompassing the nucleosome dimer, which also usually corresponds to the asymmetric unit of the crystal. As such, the two 145 bp-core nucleosomes are connected on either end by a 24 bp-linker DNA arm.

## Table 1 | Summary of linker histone-nucleosome assembly crystal structures with experimentally modeled linker histone

| Assembly | NRL[a] | Unit[b] | LH mode[c] | Resolution | PDB code |
|---|---|---|---|---|---|
| 169a-H5 | 169 bp | 2N | D/L2 | 2.84 Å | 7XVM |
| 169a-H1.0 | 169 bp | 2N | L1/L1 | 3.20 Å | 6LAB[d] |
| 169a-H1.0 | 169 bp | 2N (4N) | L1b/L1b/--/-- | 3.39 Å | 7XX6 |
| 169an-H1.0 | 169 bp | 2N (4N) | L2b/--/--/-- | 3.51 Å | 7XVL |
| 169a-H1.3 | 169 bp | 2N | F/-- | 3.19 Å | 7XX5 |
| 169a-H1x | 169 bp | 2N | L1/L1 | 2.70 Å | 8YTI |
| 338b-H1x | 169 bp | 2N* | L1/-- | 2.50 Å | 6L9Z[d] |
| 343c-H1.0 | 171.5 bp | 4N | D/D/--/-- | 3.89 Å | 6LA2[d] |
| 349c-H1.0 | 174.5 bp | Fiber (2N) | L2/D | 3.40/3.70 Å | 6LA8/ 6LA9[e] |
| 353e-H1.0 | 176.5 bp | 2N | D/D | 2.86 Å | 7COW[d] |

[a]Nucleosome repeat length within the crystal.

[b]Annealed nucleosomal unit within the crystal: paired nucleosomes (2N), paired di-nucleosomes (4N), or continuous fibers (Fiber). The 338b assembly entails a di-nucleosomal DNA fragment displaying stacking of the blunt-ended DNA termini in the crystal (2N*). These Unit values, or those in parentheses where given, correspond to the asymmetric unit of the crystal.

[c]Observed linker histone binding modes. Based on the expectation of one LH molecule bound per nucleosome, a "--" denotes the absence of a modeled LH molecule.

[d]Adhireksan et al., 2021 (ref. 21). This reference also describes the other constructs, from a set of 24 total, which either diffracted X-rays poorly or yielded data sets with a resolution of 5.5 Å or better.

[e]Adhireksan et al., 2020 (ref. 22).

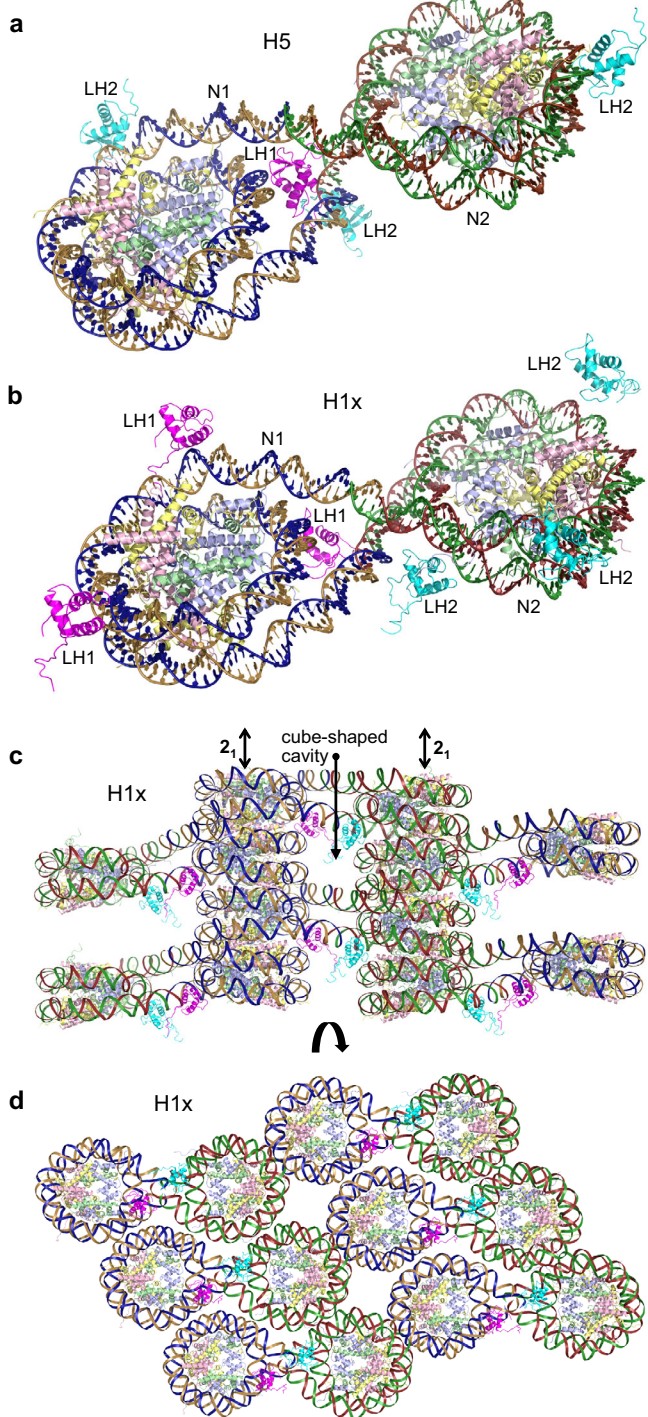

**Fig. 1 | Nucleosome-linker histone assembly structure and crystal packing.**
**a, b** Asymmetric units of the H5-169a (**a**) H1x-169a (**b**) crystals, which consist of a dimerized nucleosome pair, from annealing of the 4-nucleotide overhangs, and two LH molecules (LH1, magenta; LH2, cyan). Two (**a**) or four (**b**) additional (crystallographically identical) LH copies are shown to illustrate the full complement of LH-DNA contact interfaces that occur for each nucleosome. **c, d** Internucleosomal contacts in the crystal, viewed orthogonal to (**c**) and along (**d**) the 2-fold screw ($2_1$) axes. **a–d** The LHs and DNA strands corresponding to the two crystallographically distinct nucleosome-LH assemblies are colored differently, while the H3, H4, H2A, and H2B core histones appear in pale colors of blue, green, yellow, and pink, respectively.

The 169 nucleosome dimers are arranged in layers in the crystal, with nearly fully overlapping face-to-face stacking of the nucleosomes perpendicular to the layer planes (Fig. 1c). The face-to-face stacking from one layer to the next involves roughly opposite nucleosome orientations, such that the two nucleosomes within a dimer each interact with two different dimers from adjacent layers. This coincides with a zig-zag disposition of dimers along the stacking plane, and in the orthogonal directions, the dimers are again offset, such that the two nucleosomes of a given dimer are alternatively positioned in the recesses within and between dimers (Fig. 1d). Depending on the LH variant and crystallization outcome, the LHs can be situated either between the linker DNA sections of a given nucleosome (Fig. 1a, LH1, magenta; on-dyad mode) or within the roughly cube-shaped cavity formed by DNA boundaries from eight different nucleosomes over three consecutive layers (Fig. 1b–d; linking modes).

A total of 11 crystallographically distinct $LH_{GD}$ structures for H1x, H1.0, H1.3, and H5 from 7 different nucleosome-LH assembly crystals were solved. The nucleosome-LH assembly crystals coincide with nearly identical nucleosome packing (lattice) configurations, while the $LH_{GD}$ structures all adopt the winged helix fold and are overall similar to each other (Supplementary Fig. 5). The three α-helices (α1–α3) forming the N-terminal core module are connected by loop regions (ℓ1, ℓ2), and the C-terminal element consists of two strands (s1, s2), which fold back over the core module and interact with the C-terminal portion of ℓ1. A third loop (ℓ3) connects s1 and s2. In spite of conservation of the winged helix motif, the different $LH_{GD}$ structures display significant conformational variability in the loop regions, both between and within variants— in particular for ℓ3. This results from an overall degree of plasticity that allows accommodation of differences in binding context (see below).

### On-dyad consensus mode across nucleosomal constructs

In our earlier structural solutions of a diverse set of nucleosomal constructs, in cases where LH could be explicitly modeled, we consistently observed binding in the on-dyad mode (Fig. 2). This includes three different di-nucleosomal constructs with distinct linker DNA lengths that crystallize as closed/circular di-nucleosomes or tetra-nucleosomes, or as continuous nucleosome fibers. The $LH_{GD}$ is known to have three major DNA-binding surfaces/motifs that interface with the nucleosome dyad region and the two linker DNA arms. This entails the α2/α3/ℓ3, α1/ℓ2/α3, and ℓ1/s1/s2 motifs, which interact respectively with the central (widened) minor groove at the nucleosome dyad, a minor-and-major groove section of linker DNA arm 1, and phospho-diester elements (generally over a minor groove region) of linker DNA arm 2 (Figs. 2 and 3).

In our structures of 343c, 349c, and 353e assembled with H1.0, we observed 5 instances of on-dyad binding (of a total of 8 potential sites), which nonetheless coincide with substantial conformational differences in the linker DNA arms (Fig. 2). However, the extent of conformational variability of linker DNA arm 2 is substantially greater than that of either linker DNA arm 1 or the bound dyad region, which essentially has a fixed conformation. This is consistent with the relatively sparse number of DNA contacts associated with the ℓ1/s1/s2 interface relative to either the α2/α3/ℓ3 or α1/ℓ2/α3 interfaces (Fig. 3; Supplementary Fig. 6; Supplementary Data 1).

For the 169 construct assemblies, we also observe at least one example of on-dyad binding. We had originally tested this class of construct, having short linker DNA arms, to ensure that high-affinity LH binding to a single nucleosome can be achieved. H1x, H1.0, H1.3, and H5 all display very similar kinetics and high affinities towards the 169a nucleosome in solution. The dissociation constants ($K_D$) span a narrow range of between 15 and 45 pM (Supplementary Table 3; Supplementary Fig. 7). In the H5-169a crystal structure, on-dyad binding is observed at one of the two paired nucleosomes, although the $H5_{GD}$ assumes the orientation whereby the α1/ℓ2/α3 motif also interacts with

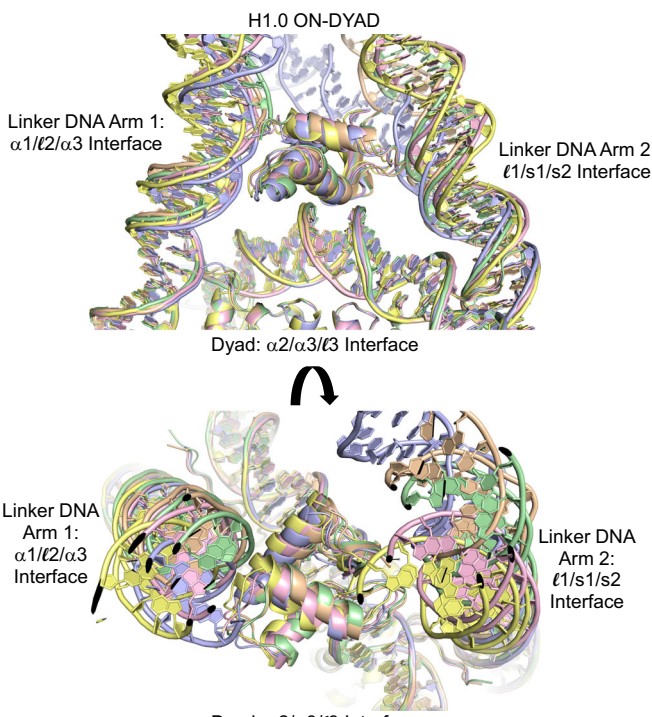

**H1.0 ON-DYAD**

Linker DNA Arm 1:
α1/ℓ2/α3 Interface

Linker DNA Arm 2:
ℓ1/s1/s2 Interface

Dyad: α2/α3/ℓ3 Interface

Linker DNA
Arm 1:
α1/ℓ2/α3
Interface

Linker DNA
Arm 2:
ℓ1/s1/s2
Interface

Dyad: α2/α3/ℓ3 Interface

**Fig. 2 | Differential contact strength and stringency of linker histone-DNA interactions between the three globular domain binding interfaces.** Five unique on-dyad models from three different H1.0-dinucleosome assembly crystal structures (*pdb* codes 6LA9 [H1.0], 6LA2 [H1.0]₂, and 7COW [H1.0]₂, corresponding to the 349c, 343c, and 353e constructs, respectively)[21,22] are superimposed and rendered with distinct colors (see also Supplementary Fig. 6). The extent of conformational variability of the second linker DNA arm is substantially greater than that of either the first linker DNA arm or the bound dyad region. This is consistent with the relatively sparse number of DNA contacts associated with the ℓ1/s1/s2 interface relative to either the α2/α3/ℓ3 or α1/ℓ2/α3 interface (see also Fig. 3).

elements of the major groove that is formed by the annealed overhangs from the paired nucleosome (Fig. 4a and b).

## Linking and on-dyad modes adopted in parallel
In the H5-169a structure, the nucleosome paired with that having the on-dyad-bound LH does not serve as a host to another instance of on-dyad binding. Instead, a second H5 molecule is associated via a linking mode, with DNA elements from four different nucleosomes, comprising the SHL 4/4′ DNA supergroove of one nucleosome (N4), the SHL 5.5 major groove of a second nucleosome (N3), and the paired linker DNA section of the neighboring nucleosome dimer (N1/N2; Fig. 4a and c). The major groove interface entails extensive DNA base contacts with C-terminal portions of the H5$_{GD}$ α1 and α2 that would suggest the potential for preferential recognition (direct readout) of the DNA sequence (Fig. 4d).

The linking mode interactions displayed by the H5$_{GD}$ are analogous to those of H1.0, which was seen linking together three separate fibers/nucleosomes by interacting with five different DNA elements, in our earlier dinucleosome (349c) crystal structures[22–24] (Figs. 3, 4e and f). The H1.0$_{GD}$ forms interactions with nucleosome core DNA elements that are configured in a fashion similar to that which supports on-dyad binding within a single nucleosome. This is apparent from structural comparison as well as by the observation that the residues involved in H-bonding with the DNA are nearly identical between the on-dyad and linking modes (H1.0$^D$ and H1.0$^{L2}$, respectively, Fig. 3 and Supplementary Data 1). In fact, the H1.0$_{GD}$-DNA contact area for the linking mode (1534 Å²; calculated as the change in solvent-accessible surface area; Supplementary Table 4) is very similar compared to that

for the on-dyad mode, being either slightly higher (1569 Å²) or lower (1425 Å²) depending on whether one includes the contribution from a close contact 'above' with the linker DNA that is shared between the N + 1 and N + 2 nucleosomes (Fig. 4e).

In the H5-169a structure, the GD-DNA contact areas are also very similar between the on-dyad (2199 Å²) and linking (2188 Å²) modes (Supplementary Table 4). Moreover, the H1.0-349c and H5-169a linking modes share application of the α2/α3/ℓ3 and ℓ1/s1/s2 motifs in a manner similar to the corresponding on-dyad modes, as well as engage additional GD residues to achieve contacts with DNA elements outside of the on-dyad binding-triad (Figs. 3 and 4 and Supplementary Data 1). In fact, in either case, the α2/α3/ℓ3 motif is associated with a widened minor groove nucleosome core element, which− as a consequence of histone-mediated bending into the (inward-facing) major groove− is nearly identical in structure to that at the dyad (nucleosome center; Fig. 4g and h). On the other hand, while the α1/ℓ2/α3 motif is seen to recognize a similar minor-and-major groove element for the H1.0-349c on-dyad and linking modes and the H5-169a on-dyad mode, in the case of the H5-169a linking mode, it is only partially utilized and in a split fashion. More specifically, the central region of the H5 α1/ℓ2/α3 surface remains unengaged, while the peripheral portions associate with two separate DNA elements (Fig. 4h). In this way, a novel interface is created, whereby the α1/ℓ1/α2 surface binds to an extended major groove section with shape/electrostatic base-complementarity (Fig. 5a). This fourth interaction surface of the GD allows for a tetrahedral-like DNA binding sphere, as it is also observed for the H1.0-349c on-dyad mode, where it is utilized (albeit only sparingly) to contact a neighboring section of linker DNA (Fig. 4e). Overall, the H1.0 and H5 GDs preside over DNA binding elements beyond the canonical dyad and linker DNA arm 1 and 2 motifs that include both electropositive and electronegative patches (Figs. 3, 5a and b). In spite of sharing >80% sequence identity, the H1.0$_{GD}$ and H5$_{GD}$ display distinctions in charge distribution, amongst other features, which likely contribute to differences in DNA binding preferences (see also below).

## Variant-dependency in linking mode binding
The ability of H5 to associate with the 169a nucleosome dimer via the on-dyad mode indicates that this LH binding mode is permissible within 169 crystals, in spite of structural constraints that may be imposed by the nature of nucleosome pairing. Nonetheless, the absence of on-dyad H5 binding in the second, opposing, nucleosome of the dimer may be a consequence of either a special favorability associated with the observed H5 occupancy at the linking-mode binding site or an on-dyad-disfavoring/incompatible nucleosome conformation (although a significant degree of plasticity is inherent in the nature of on-dyad LH$_{GD}$-nucleosome interactions; Fig. 2). In any case, for the remainder of the LH-169 assembly crystals analyzed, no further instances of on-dyad binding were observed (Fig. 5c–h).

For the H1x, H1.0, and H1.3 variants, we analyzed many data sets across a variety of nucleosomal constructs. Crystals of H1x-169a consistently displayed a type of linking mode binding that is distinct from that described above for H1.0 and H5. Although interactions are formed between the H1x$_{GD}$ and three nucleosomes (four DNA sections), there is only a single extensive interface, which involves the α1/ℓ2/α3 motif (Fig. 5c and d). This interface accounts for the majority of the contact area (819 Å² of the 1209 Å² total; Supplementary Table 4), while the three other contact points encompass sparse interactions. The main interface involves positioning of α3 within the minor groove around SHL 6, accompanied by concerted binding of the N-terminal region of α1 to the DNA backbone (Fig. 6). This interface in its entirety spans over a turn of the double helix as N-terminal (and in one H1x copy also C-terminal) residues emerging from the GD track further along the DNA to provide supporting interactions (Supplementary Data 1; Fig. 6a). The result is a pronounced conformational change entailing a widened minor groove, which permits extensive hydrogen

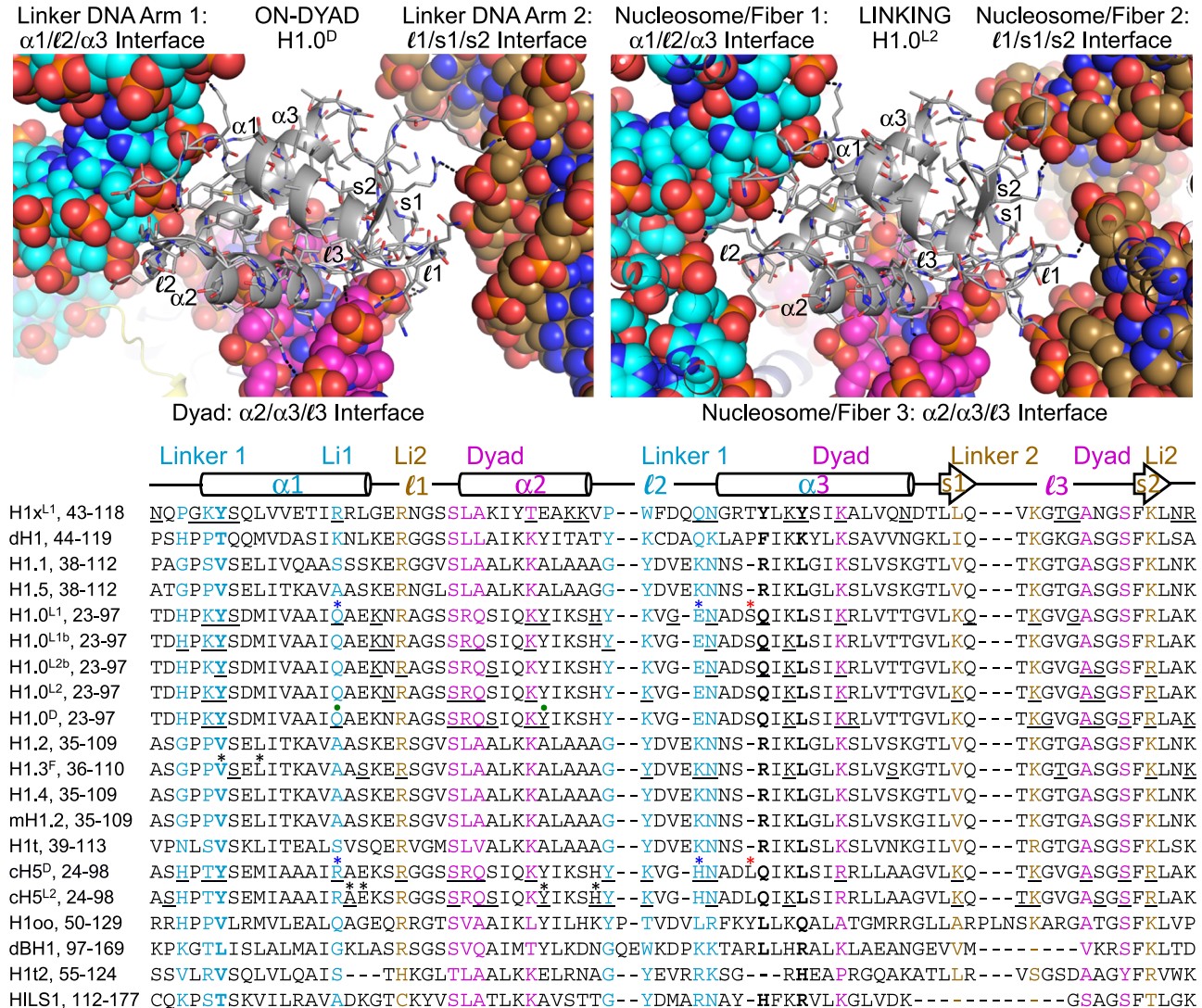

**Fig. 3 | Summary of linker histone-nucleosome interactions and sequence homology comparison.** A sequence alignment of 15 different LH types with respect to the H1x$_{GD}$ (residues 45–116) is shown in descending order of homology. Examples of on-dyad (D) and linking mode (L2) binding from the 349c dinucleosome fiber structure are shown above (*pdb* codes 6LA9 and 6LA8)[22,24]. All variants are *H. sapiens*, with the exception of *Drosophila*, mouse, and chicken (d, m, and c prefixes, respectively). DNA-hydrogen bonding residues (Supplementary Data 1) are underlined and colored relative to those of H5 bound in the on-dyad mode (interactions with nucleosome dyad, linker DNA arm 1, and linker DNA arm 2 in magenta, cyan, and brown, respectively). Superscripts denote nucleosome association mode (D, on-dyad; F, off-dyad; linking modes, L1/L1b/L2/L2b). The H1x tyrosine triad involved in the L1 binding mode and corresponding residues in the other variants are shown in bold. Blue/red asterisks designate amino acid differences between H1.0 and H5 that appear to influence the proclivity of these two closely related variants for adopting the on-dyad mode (blue; green dots, internucleosomal H-bonding residues[22,24]) versus the L1 mode (red). Black asterisks denote residues in H1.3 involved in hydrophobic contacts with thymine bases in the major groove and residues in H5 recognizing the major groove sequence element CGG/CCG.

bonding and hydrophobic contacts with the sugar-phosphate backbone as well as a hydrogen bond between the N3 ring nitrogen atom of a guanine base and the penetrating side chain of Y91. Nonetheless, this SHL 6 interaction alone is not sufficient to allow specific association with the corresponding NCP, and therefore the presence of linker DNA is essential for specific H1x binding to individual nucleosomes in solution (Supplementary Fig. 8). Hydroxyl radical footprinting analysis of the 169a-LH assembly indicates that H1x interacts with both dyad and linker DNA regions but not SHL ± 6 (Supplementary Fig. 9), which is consistent with H1x binding to mononucleosomes in the on-dyad mode, as observed by cryo-EM[25].

The nucleosome context-dependent binding of H1x— evident from the on-dyad mode of association to individual (169a) nucleosomes in solution versus the type of linking mode adopted in the dense multi-nucleosomal environment in the crystals— indicates a context

sensitivity that is also seen in a 2.5 Å resolution structure of H1x assembled with a 338 bp-dinucleosome (338b)[21]. This construct was engineered to mimic the nucleosome-LH dimer configuration of the 169 crystals (Figs. 1 and 5c), and the ensuing dinucleosome crystal packing creates a similar environment (Supplementary Fig. 10). However, only one H1x molecule per dinucleosome can be resolved, although it binds in the identical linking mode as seen in the H1x-169a structure. The absence of a second H1x copy binding in a side-by-side fashion coincides with a shift in the disposition of the corresponding interacting nucleosomes, which would preclude association in the same fashion (Supplementary Fig. 10).

In screening through many H1.0-169 data sets, we discovered three different crystal types where H1.0 assumes distinct binding modes. One of these coincides with the same linking mode seen in the H1x-169a crystals. The GD in the H1.0-169a structure interacts with the

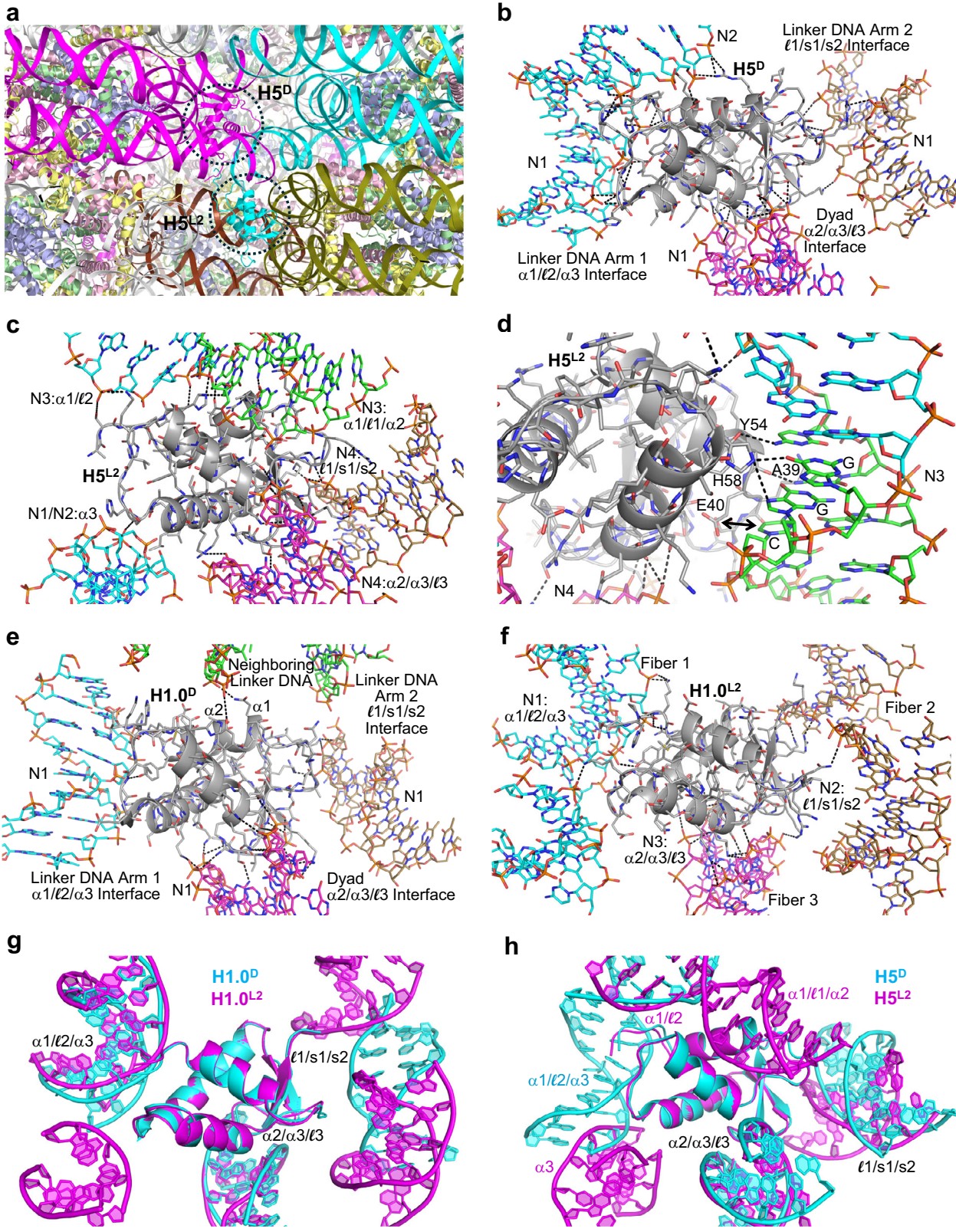

four surrounding DNA elements in nearly identical fashion as H1x (Fig. 5e), although this is achieved with many differences in amino acid sequence between the two variants for the interfacial residues (Fig. 3; Supplementary Data 1). Like H1x, most of the H1.0$_{GD}$-DNA interfacial area (783 Å$^2$ of the 1336 Å$^2$ total; Supplementary Table 4) stems from the α1/ℓ2/α3 motif interaction with the SHL 6 widened minor groove.

Given the common character of the single-major-interface LH$_{GD}$ binding mode in these H1x- and H1.0-169a structures (where the α1/ℓ2/α3 motif is the apparent driver), we assign this as a linking mode 1 (**L1**) category, with which to distinguish between the other linking mode configurations (Fig. 7). This can be contrasted with the H1.0 and H5 linking modes outlined in the previous sections, whereby the overall commonality is the presence of two major interfaces that involve both

**Fig. 4 | Similarities and distinctions between on-dyad and linking modes for the sequence-homologous H5 and H1.0 linker histones. a** Section of H5-169a crystal showing the nucleosome binding environment for the two H5 molecules, which are associated in the on-dyad (D, magenta) and linking (L2, cyan) modes. The DNA of the four interacting nucleosomes is colored individually, and the H3, H4, H2A, and H2B core histones appear in pale colors of blue, green, yellow, and pink, respectively. **b–f** Details of LH$_{GD}$-DNA interactions for H5 (**b–d**; H5-169a) and H1.0 (**e, f**; from the 349c dinucleosome fiber structure[22,24]). DNA segments that are in close proximity to the LH$_{GD}$ (grey C atoms) are shown with C atoms colored relative to the identity of interaction with LH motifs (α1/ℓ2/α3, cyan; α2/α3/ℓ3, magenta; ℓ1/s1/s2, brown; alternative motifs/elements, green). Hydrogen bonds are shown as dashed lines. Panel **d** details the engagement of the C-terminal α1/ℓ1/α2 surface, providing shape, electrostatic (carboxylate group [E40] contact with the electropositive face of cytosine, arrow), and hydrogen bonding complementarity towards the major groove CGG element. **g, h** Superimpositions, with respect to the LH$_{GD}$, of the on-dyad (D) and linking (L2) mode interaction regions shown in (**e**) and (**f**) for H1.0 (**g**; neighboring linker DNA [H1.0$^D$] not shown for clarity) and (**b**) and (**c**) for H5 (**h**).

the α1/ℓ2/α3 and the α2/α3/ℓ3 motifs, or in the case of the H5-169a structure, α2/α3/ℓ3 together with an alternative LH$_{GD}$ motif. As such, we assign this as the **L2** category, which is more closely analogous to the on-dyad binding mode that is characterized by two major DNA interfaces based on the α1/ℓ2/α3 and the α2/α3/ℓ3 motifs.

Regarding the other two H1.0 binding modes observed in 169 crystals, one corresponds to a crystal growth polymorph of the H1.0-169a assembly that coincides with a type of binding referred to as L1b, since it appears as a variant of the L1 mode (Figs. 5f and 7a; Supplementary Fig. 11). In either case, an α1/ℓ2/α3-DNA minor groove interaction is the major interface that is supported with otherwise sparse contacts to several surrounding DNA elements. However, for the L1b mode, the engagement of the α1/ℓ2/α3 motif is with an SHL 4 (as opposed to SHL 6 in the L1 mode) minor groove and coincides with a less invasive interaction that results in a significantly reduced total interfacial area (849 Å$^2$ [L1b] versus 1336 Å$^2$ [L1]; Supplementary Table 4). A distinct linking mode example, designated as L2b, was also obtained for crystals of H1.0 assembled with the 169an construct (Figs. 5g and 7b). It is assigned to the L2 class, as it has the analogous profile as seen for the L2 mode of H1.0-349c crystals, by virtue of having two major interfaces based on the α1/ℓ2/α3 and the α2/α3/ℓ3 motifs together with additional sparse contacts to surrounding DNA elements. Indeed, the total interfacial areas for the two systems (1366 Å$^2$ [169an] versus 1534 Å$^2$ [349c]; Supplementary Table 4) are both similar and larger than those seen for the L1/L1b examples (Fig. 7c).

In contrast to the binding displayed by H1x and H1.0, the 169a structure obtained for H1.3 shows an off-dyad type mode, in which the H1.3$_{GD}$ is situated over SHL 0.5 (directly adjacent to the dyad, i.e., SHL 0), where it makes contacts with either end of the major groove (Fig. 5h). In spite of this off-the-dyad positioning, it is still able to form interactions with both linker DNA sections, but with contacts to one arm (comprising 982 Å$^2$ of the 1562 Å$^2$ total interfacial area; Supplementary Table 4) being significantly more extensive than the other. The extensively contacted linker DNA arm involves α1/ℓ2/α3 elements in a manner similar to that seen for on-dyad binding, such as in the H5-169a structure. However, for H1.3, the α1 N-terminus is inserted into the major groove, where hydrophobic contacts to 3 thymine bases of the poly-A/T track are formed with a leucine and valine side chain. The nature of these contacts suggests that the DNA sequence could play a direct role in LH binding, and the corresponding V$_{41}$SEL$_{44}$ sequence motif is in fact unique to the H1.1–H1.5 variant subfamily (Fig. 3).

## Nucleosome core double helix remodeling
In the L1 mode H1x- and H1.0-169a structures, both of the shared linker DNA sections are relatively straight, until around SHL 6, at which point there is a sharp kink of about 40° in each nucleosome (Fig. 6d). This differs from the respective symmetry-related halves (SHL 6′), where there is only about a 20° bend from histone octamer association (Fig. 6c). The distinct 40° kinks, which are directed into the major groove, are a consequence of the invasive α1/ℓ2/α3-nucleosome (N1) interactions (Figs. 5d, e and 6). The insertion of the LH α3 helix into the minor groove and the associated supporting motifs result in remodeling the double helix around SHL 6 to have a pronounced bulge with substantial A-form-like character (Fig. 6a). In particular, a triad of tyrosine residues stack against each other in H1x– which are correspondingly a tyrosine, glutamine, and a leucine residue in H1.0 (Fig. 3)– and in either case these three residues stack against the exposed face of the splayed-open minor groove, with two adjacent nucleotides adopting C3′-endo (A-form) sugar pucker. The DNA structural remodeling renders the double helix with closer contacts to histone octamer elements around the nucleosome core terminus (SHL 6.5; Fig. 6b).

## Linker histones link together multiple nucleosome fibers
Given our earlier observation that H1.0 binds in a linking (L2) mode by interacting with multiple nucleosome fibers in dinucleosome crystals[22–24], in conjunction with the above results, we conducted LH binding experiments with condensates generated in vitro. The condensates, which are formed under conditions that promote self-association of nucleosomal arrays, have been shown to provide an excellent model system for studying the condensed chromatin environment present in interphase chromosomes[26,27]. When varying amounts of LH are added to preformed condensates, binding to nucleosomes does not cease at unitary (1:1 LH:nucleosome) stoichiometry but instead LH continues to associate with the condensates up to and potentially beyond 6:1 added H1 ratios (Fig. 8a). The binding profiles and points of plateau differ between H1.0 and H1x, which moreover display distinct saturation behavior when the amount of bound LH in the arrays is quantified (Supplementary Figs. 12 and 13). For H1.0, association levels approach nearly 3 LH molecules bound per nucleosome and do not appear to have achieved saturation by 6:1 added LH. In contrast, for H1x, the condensates harbor binding sites for up to about 4 molecules per nucleosome, which appears to also correspond to a saturation point. Analogous behavior, with variant-specific differences in loading and saturation profiles, are seen for the DNA replication-dependent members (H1.1–5 variants) of the somatic LH family (Supplementary Fig. 14). This indicates that there are multiple LH binding sites within condensed nucleosome fiber assemblies, and at least some of these binding motifs are occupied in a variant-dependent fashion.

We also conducted microscopic imaging of condensates with arrays that had been fluorescently labeled (Fig. 8b and c; Supplementary Figs. 15 and 16). In the absence of LH or in the presence of LH at low stoichiometry, different condensates have a nearly identical appearance, entailing a typically spherical geometry with a diameter of 260 ± 100 nm. However, at higher LH stoichiometry, the spherical condensates are seen to aggregate into much larger assemblies resembling beads on a string in varying states of compaction. The effect is apparent already at around 1:1 LH:nucleosome stoichiometry for some of the H1.1–5 variants and for H1.0 (Fig. 8c; Supplementary Figs. 15 and 16), which displays overall greater condensate-linking activity relative to H1x, but in any case, the extent of aggregation increases with the amount of added LH. These observations indicate that H1 is able link multiple nucleosome fibers together by mediating complex internucleosomal contacts, consistent with our body of crystal structures.

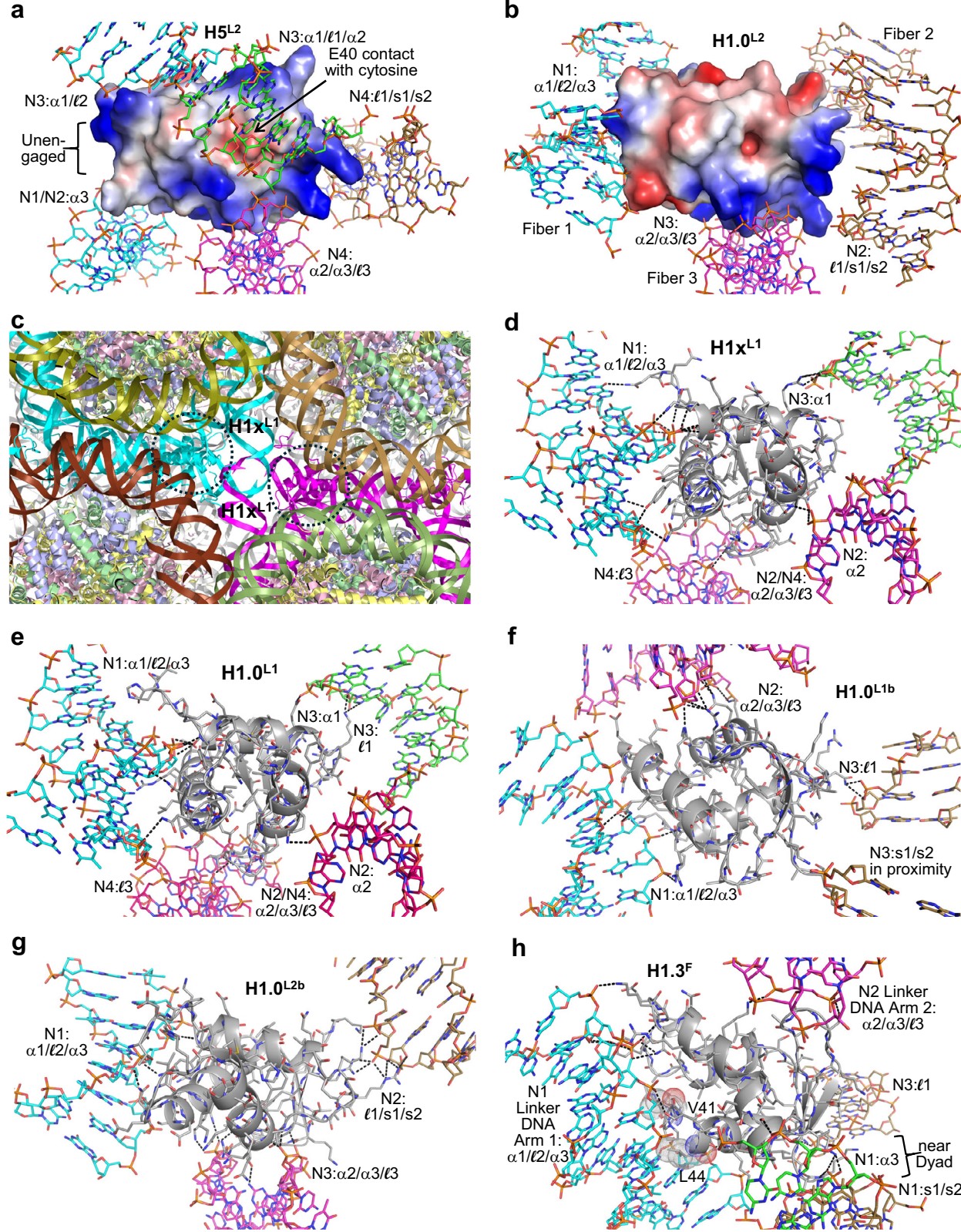

## Discussion

Previous studies have largely come to the conclusion that LHs associate with an individual nucleosome, but here and in our recent work, we explored a variety of substrates in the condensed state that reveal a proclivity for LHs to bind to multiple nucleosomes simultaneously[21,22]. Nevertheless, we also consistently observe on-dyad binding, with H5 in the context of the 169 nucleosome dimer and with

H1.0 in the context of different cohesive-ended dinucleosome constructs that form either continuous fibers, dinucleosome dimers, or closed dinucleosomes. This coincides with the same binding mode as observed before in crystallographic and cryo-EM investigations[25,28,29], wherein a significant degree of binding plasticity is apparent from the ability of the LH$_{GD}$ to associate with a variety of nucleosome conformations (Fig. 2; Supplementary Figs. 6 and 17), while the exact

**Fig. 5 | Diversity and variant-dependency in linker histone binding. a, b** Distinctions in electrostatic profiles between the H5 and H1.0 GDs, shown in the context of the linking mode (L2) LH$_{GD}$-DNA interactions in the H5-169a (**a**) and the H1.0-349c dinucleosome fiber structures[22,24] (**b**). **c** Section of H1x-169a crystal showing the nucleosome binding environment for the two H1x molecules (cyan and magenta), which are both associated in the linking (L1) mode. The DNA of the six interacting nucleosomes is colored individually, and the H3, H4, H2A and H2B core histones appear in pale colors of blue, green, yellow, and pink, respectively. **d**−**h** Details of LH$_{GD}$-DNA (LH$_{GD}$, grey C atoms) interactions for H1x (169a), H1.0

(169a, 169a [polymorph], and 169an), and H1.3 (169a) associated, respectively, in the linking (L1, **d** and **e**; L1b, **f**; L2b, **g**) and off-dyad (F, **h**) modes. Hydrogen bonds are shown as dashed lines. **a, b, d**−**h** DNA segments that are in close proximity to the LH$_{GD}$ are shown with C atoms colored relative to the identity of interaction with LH motifs (α1/ℓ2/α3, cyan; α2/α3/ℓ3, magenta; ℓ1/s1/s2, brown; alternative motifs/elements, green). **h** The valine and leucine residues of the H1.3 V$_{41}$SEL$_{44}$ element are highlighted with space-filling rendering to illustrate contacts with thymine methyl groups over the major groove face of the poly A|T tract.

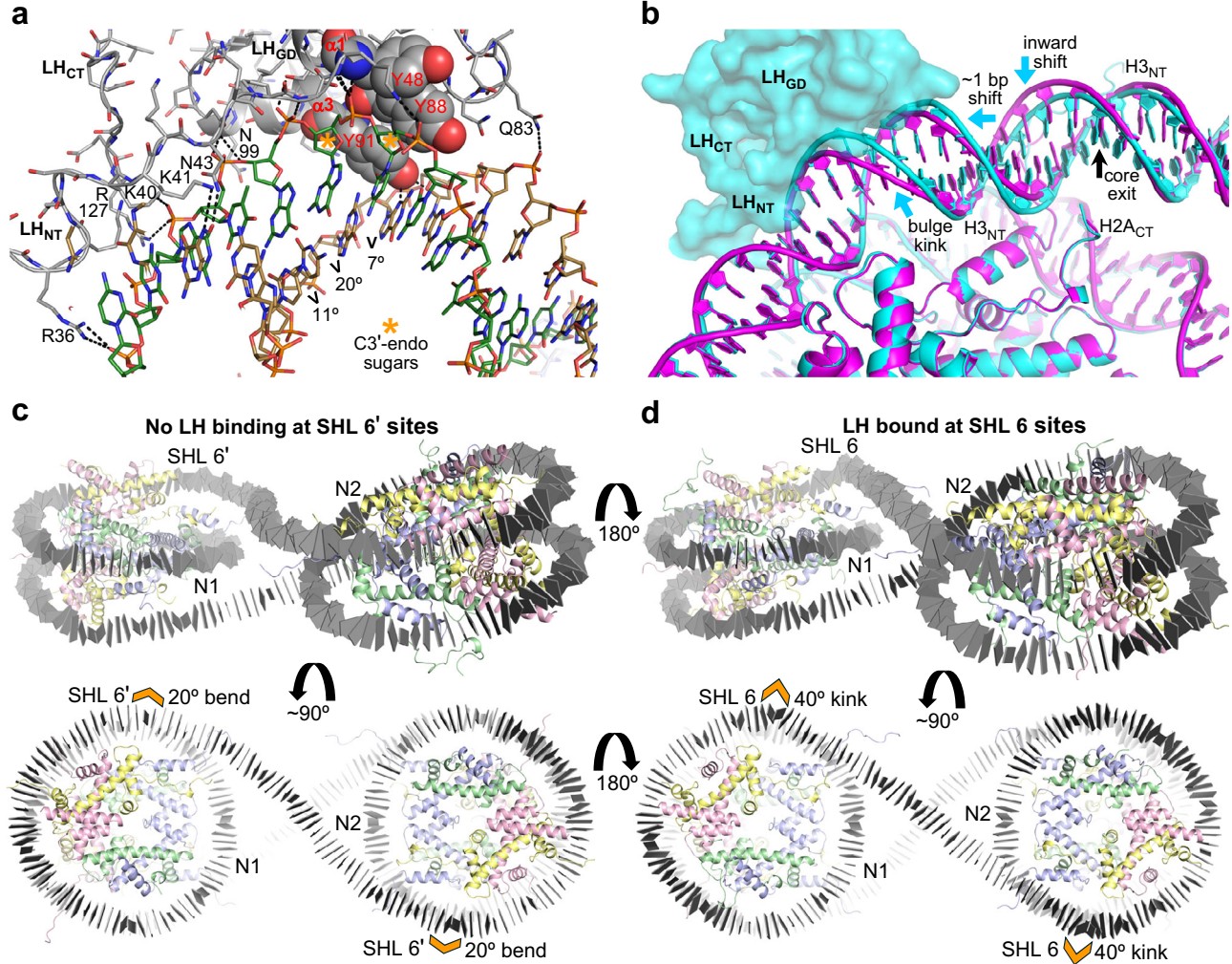

**Fig. 6 | H1x binding in the linking mode remodels nucleosome core structure. a**−**d** H1x binding in the 169a crystal structure. **a** Details of the H1x interaction over SHL 6, encompassing the LH$_{GD}$ α1/ℓ2/α3 motif binding in the DNA minor groove (the major interface in the L1 mode). The tyrosine triad is shown in space filling, and dashed lines indicate hydrogen bonding (Supplementary Data 1). Two nucleotides in contact with the Tyr triad adopt the A-form sugar pucker (asterisks). Degree values, for dinucleotide (bp step) roll, indicate sustained DNA kinking over the CCGG element. **b** Comparison of the H1x-bound SHL 6 region (cyan) with that of the

LH-free, symmetry-related SHL 6′ region (magenta). H1x binding introduces a bulge/kink into the double helix, which yields an approximate 1 bp shift in the DNA-histone octamer register and accompanying alterations in DNA-core histone proximity. The nucleosome core entry/exit point is indicated with a black arrow. **c, d** Summary of H1x binding and impact on nucleosome structure. DNA base pairs are represented as grey tiles. The pronounced kinks at the SHL 6 locations are a result of the invasive H1x interactions, which occur on only one face of the 169a nucleosome dimer (**d**).

character of on-dyad binding is further modulated by variant-specific sequence differences.

In conjunction with an alternative, 3-nucleosome/fiber-H1.0$_{GD}$, (linking, L2) binding mode seen in crystals of nucleosome fibers[22], we find here that LHs associate in a variety of ways in a context- and variant-dependent manner that includes at least several related modes in which the LH$_{GD}$ links multiple nucleosomes/fibers together, thereby providing a direct mechanism for stabilizing compact states of

chromatin. The binding diversity and variant-dependence seen in crystals are in agreement with the LH binding behavior observed for the nucleosome array condensates, both of which support multimodal and internucleosomal association profiles. Our findings suggest that LHs associate with DNA-enshrouded niches in chromatin that support favorable contacts with motifs derived from combinations of the three canonical GD binding faces (α2/α3/ℓ3, α1/ℓ2/α3, and ℓ1/s1/s2) and additional surface features that are present in a variant-specific

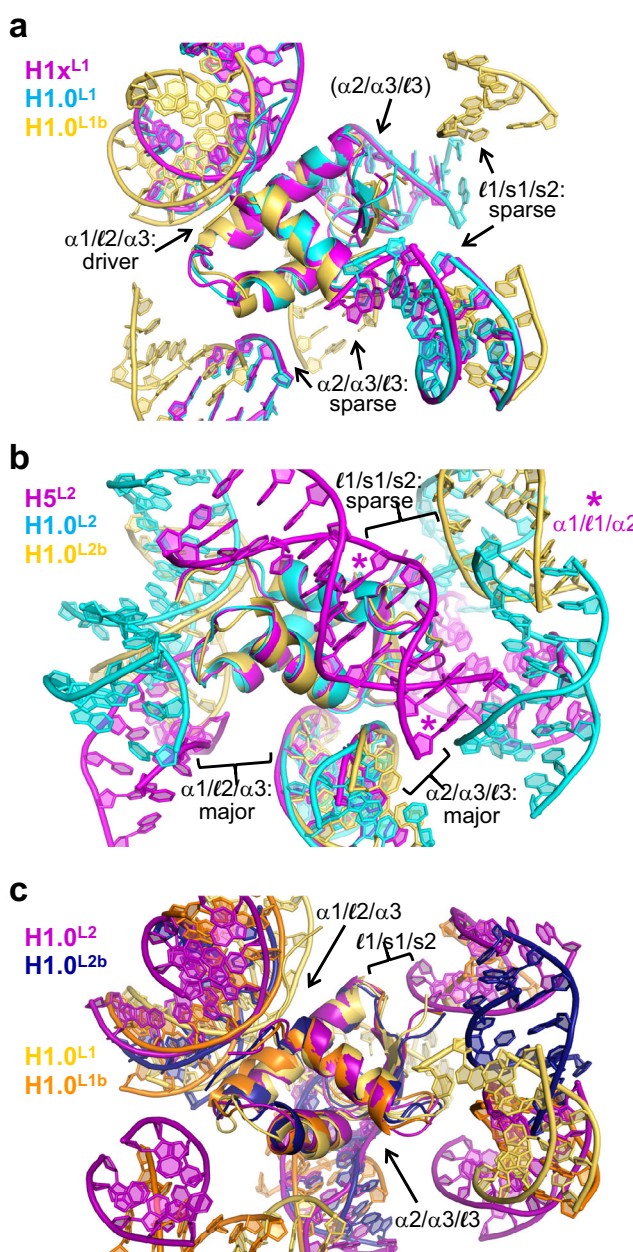

**Fig. 7 | Linker histone association with nucleosomes displays two distinct classes of linking mode binding.** Superimpositions, with respect to the LH$_{GD}$, illustrate common features associated with the L1 (**a**) and L2 (**b**) binding mode classifications and distinctions between them (**c**).

interactions with the DNA. However, the vast majority of LH tail residues could not be modeled, and yet the large LH$_{CT}$ provides a well-established driving force towards chromatin binding and compaction[10–13]. The LH tails have also evaded explicit modeling in structural studies to date, with the exception of a recent 3.6-Å resolution cryo-EM structure of 2-start, zig-zig 30 nm fibers assembled with H5 (glutaraldehyde-fixed 12-nucleosome arrays)[16]. The H5 tail domains adopt conformations rich in secondary structure and are seen to support off-dyad binding of the GD by interacting with the same as well as adjacent nucleosomes. This is consistent with the LH$_{CT}$ being an intrinsically disordered domain that undergoes a disorder-to-order transition or condensation upon binding nucleosomes, as characterized previously for H1.0[32]. Moreover, a recent cryo-EM study of LH binding to mononucleosomes (H1.0, H1x, H1.4, and a chimeric LH; nucleosome in complex with histone-binding antibodies) showed that the variant identity of the tails influences the conformation of the linker DNA arms. In either of these two detailed works, however, the LH tails are consistently supporting a single GD binding mode (off-dyad for the 30 nm fiber versus on-dyad for the mononucleosomes) that is dictated/favored by the nucleosomal context, which is in agreement with our analysis here. This suggests that the GD generally imparts structural specificity that selects for a particular LH binding mode/location, which is supported through variable LH$_{NT}$ and LH$_{CT}$ association. In any case, we envision that the LH$_{NT}$ and LH$_{CT}$ tails, like the LH$_{GD}$, are also involved in linking together multiple nucleosomes/fiber sections in chromatin, given the appropriate context.

A chromatin model based on multiple LH binding modes would help rationalize several long-standing observations. One is simply the fervent chromatin compacting activity of the LHs, which is difficult to account for with a purely mononucleosomal association dynamic. Moreover, the fact that some cell types may have an overall LH:nucleosome stoichiometry greater than unity[33] and additionally that LHs are enriched at specific loci[34,35] can be explained by nucleosome binding multiplicity, at least over certain genomic locations. In fact, the ability of LHs to bridge together multiple DNA fragments was described in the 1980s[36,37], while the capacity of a single LH molecule to link together multiple nucleosomes is consistent with super-resolution nanoscopy imaging showing that condensed chromatin features in the cell coincide with small clusters (clutches) of, on average, just several nucleosomes, but the largest and most densely compacted clutches (consisting of the greatest number of nucleosomes) are decisively enriched in LH content[30]. A more recent study found that the chromatin polymer is, in fact, arranged in a variety of packing configurations as a disordered 5- to 24-nm curvilinear chain, but again the most compact chromatin elements generally coincide with small clusters of interacting nucleosomes[31]. Furthermore, live cell imaging studies suggest there are multiple LH binding modes, with distinct pools of protein having short versus long residence times in chromatin[38–40]. This would be consistent with our findings, given two different classifications for LH binding, where one is substantially more stably associated with chromatin. We can tentatively propose that LH$_{GD}$ association with individual nucleosomes (on-dyad mode) versus binding within clusters of nucleosomes (linking modes) coincide, respectively, with the more versus less stable binding and correspondingly distinct residence times in chromatin. In this sense, we envision that the on-dyad binding sites would tend to become occupied first, followed by the linking mode sites at higher LH stoichiometry/activity, where a synergy may ensue between LH loading and chromatin compaction by creating additional linking mode sites (Fig. 9).

The nucleosome array condensate results, which show that all 7 of the somatic LH variants are capable of high (~2–4 LH:nucleosome) stoichiometry binding in vitro, are consistent with the capacity of LHs to associate with nucleosomes in >1 stoichiometry, based on the quantification of isolated cellular chromatin[33,41]. The condensate data

fashion. We envision the repertoire of binding niches available in condensed/compact chromatin to be analogous to that of a pharmacophore, but with the LH$_{GD}$ serving as the ligand that requires certain 3D arrangements of double helical elements to support high affinity (stable) binding. From our analysis here, such 'linker-histonophores' would appear to be variant-specific and dominated by the α2/α3/ℓ3 and α1/ℓ2/α3 binding faces of the GD. In fact, the heterogenous nature of local nucleosome organization observed in cellular chromatin[30,31] might suggest the existence of many distinct linker-histonophores, and thus linking mode variations (that is, potentially well beyond what we have observed so far).

For some of the LH-nucleosome assembly structures, we were able to model limited components of the LH$_{NT}$ and LH$_{CT}$ domains as they emerge from the GD, where they are seen to provide supporting

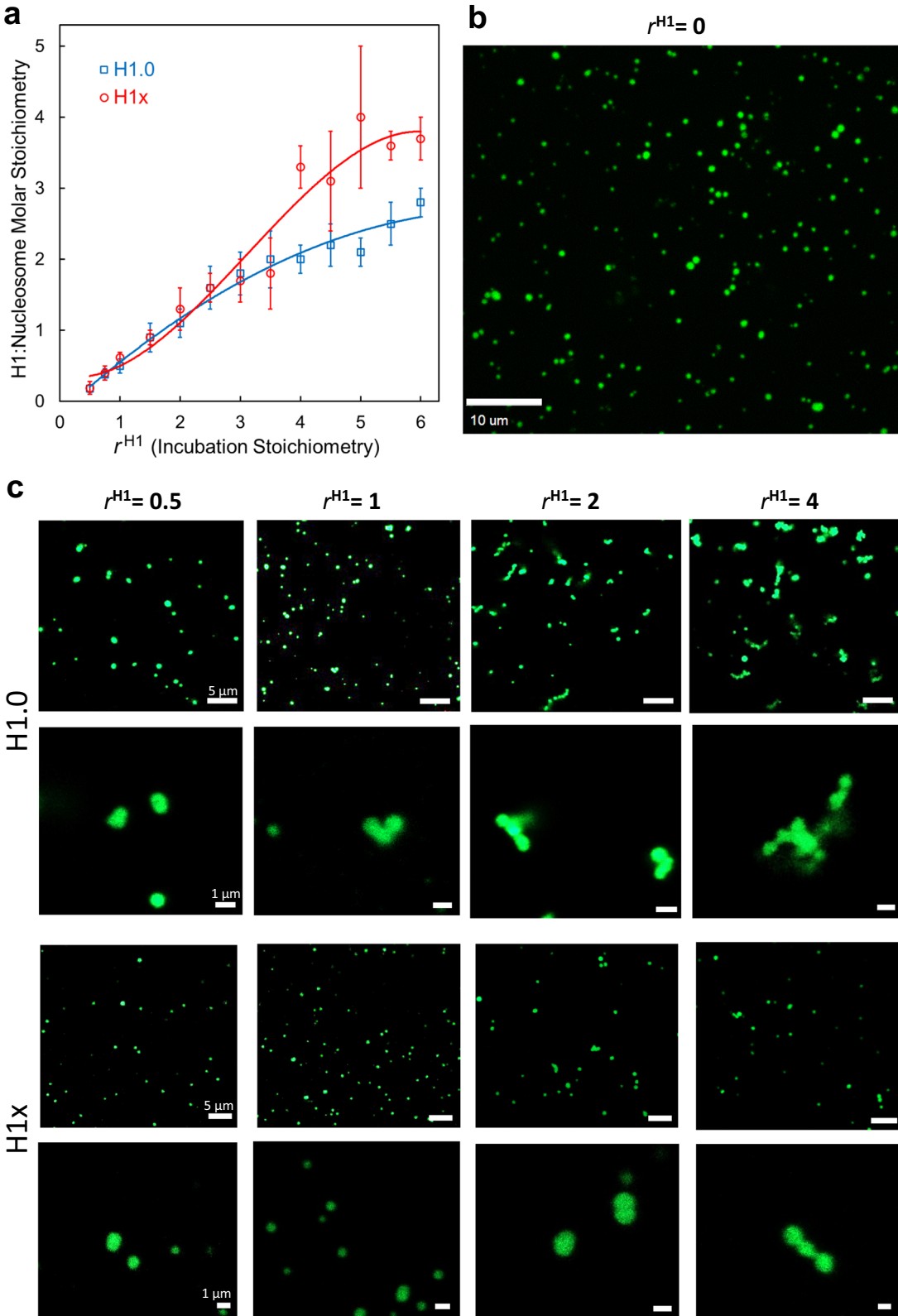

**Fig. 8 | Linker histones bind multivalently to nucleosome array condensates and stabilize complex fiber-fiber interactions. a** Condensate binding curves for H1x and H1.0. The x-axis coincides with the H1:nucleosome molar stoichiometry of H1 incubated with the array ($r^{H1}$), and the y-axis corresponds to the H1:nucleosome molar stoichiometry quantified for the amount of LH bound in the array condensates (error bars represent standard deviation, $n \geq 3$ biologically independent replicates; Supplementary Figs. 12 & 13). **b, c** Fluorescence microscopy imaging of array condensates incubated with either no LH (**b**) or varying H1:nucleosome molar stoichiometry ($r^{H1}$) of H1.0 or H1x (**c**; $n = 3$ biologically independent replicates). Source data are provided as a Source Data file.

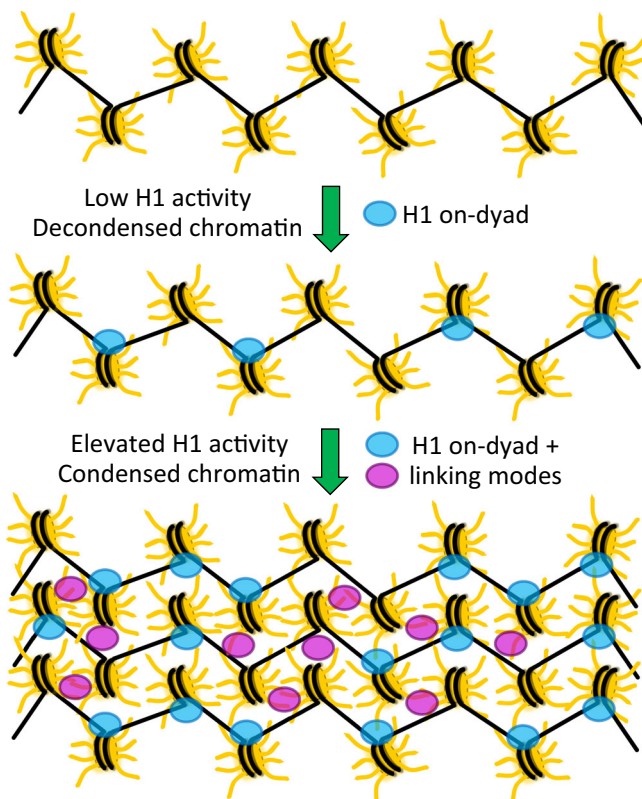

**Fig. 9 | A model for linker histone-nucleosome binding and chromatin compacting activity.** Chromatin fiber sections are represented schematically, with yellow core histones, black DNA, and cyan/magenta H1.

lend support to this in vivo LH activity and moreover, demonstrate potential chromatin compacting/condensing consequences that are consistent with the characteristics illustrated by the crystal structures. While the biological prevalence of multi-nucleosome LH binding remains unclear, an LH:nucleosome stoichiometry greater than unity is not, a priori, a prerequisite for the in vivo relevance of linking-mode LH binding. In fact, a recent cryo-EM analysis of tetra-nucleosome arrays, having different and physiologically relevant nucleosome repeat lengths (NRLs), showed that certain NRLs are refractory to H1 binding, even under saturating conditions (H1:nucleosome molar stoichiometry of 1.2)[42]. This suggests that a subset of nucleosomes in the cell is not receptive to on/off-dyad H1 binding, which would effectively increase the pool of available H1. In any case, the main argument here is that a > 1 bound H1:nucleosome stoichiometry would necessitate additional LH-nucleosome interaction types (i.e., linking modes) beyond the conventional on/off-dyad modes. As to why examples of LH binding in the linking mode had not been observed prior to the first crystal structure report using our designed nucleosomal system[22], we infer that this required a substrate that is capable of both supporting and revealing this type of LH association. The scarcity of prior crystal structures for this key chromatin complex[28,29] may be a reflection of the prevalence of LH binding diversity/multiplicity, which would tend to thwart crystallization efforts. This was initially our motivation for engineering nucleosomal constructs that could promote lattice order by replacing the loosely stacking blunt-ended DNA termini with annealing single-stranded overhangs[21,22]. Conversely, solution-state-based methods, like single particle reconstructions (cryo-EM) and footprinting analysis of LH-nucleosome assemblies, generally employ very low ionic strength and divalent metal-free conditions to prevent nucleosome aggregation, which correspondingly selects against linking mode LH binding.

Although there are 11 variants in mammals, LHs have frequently been treated as a single protein type, H1. However, in recent years, there has been more interest in characterizing the functional specialization of each of the variants. Variant-specific activities span distinctions in chromatin affinity, chromatin compacting ability, protein interaction profile, and genomic localization as well as gene- and transaction-dependent function[5]. These activities are moreover modulated by a variety of post-translational modifications at both the GD and tail sites of the LHs. The distinctions we find between H1x, H1.0, H1.3, and the avian version of H1.0 (H5) are a reflection of their different nucleosome association predispositions, and this could help rationalize variant-dependent differences observed in chromatin binding and genomic activity. We also obtained well-diffracting crystals of different constructs assembled with H1.1, H1.2, H1.4, and H1.5. However, in all these cases, as well as in many instances with the four other somatic variants, LH-associated electron density was either absent or too poorly defined for atomic modeling[21,22]. Nonetheless, we interpret this as further evidence for LH binding diversity, in terms of what is likely both static and dynamic disorder, as well as variant-dependent behavior. For instance, the proclivity of H1x to associate in the one particular linking mode (L1) is at least partly attributable to the tryrosine-triad of the α1/α3 motif, which is a sequence feature unique to H1x (Fig. 3) that provides a prominent binding surface for minor groove insertion and nucleosome core deformation. H1.0 is the only other mammalian LH variant having at least one of these tyrosine residues (Y28), and it is also coincides with the only other L1 mode example. In this sense, one can flag residue differences that contribute to the proclivity of H1x and H1.0 for binding in the L1 mode and those that would enhance the tendency of H5 to adopt the on-dyad mode relative to its closest human homolog, H1.0 (Fig. 3). Considering how H1.0 is seen to associate in at least four different linking mode variations, it is interesting to speculate that this could be connected to a specialized function of H1.0 in promoting widespread heterochromatin formation in terminally differentiated cells, where it is most prevalent[43].

In addition to the differential preferences apparent between variants with respect to DNA binding site configuration, H5 and H1.3 also display interactions suggestive of DNA sequence-dependent association— a phenomenon observed in earlier LH studies[44]. Collectively, the variant-dependent behavior characterized here implies that local chromatin architecture-based discrimination contributes to the specialization of LH family members by modulating chromatin affinity, compaction, and localization attributes. With the potential advantage of mimicking the solid-like attributes of condensed/compact chromatin, future crystallographic studies could shed light on properties of other architectural factors as well as the possible roles of post-translational modifications in modulating chromatin binding and compaction.

## Methods

### Crystallization and data collection

*H. sapiens* core histones and LHs, as well as DNA fragments and chicken H5, were overexpressed individually in *E. coli* and processed/purified, followed by core histone octamer refolding and nucleosome reconstitution[21,22]. The DNA fragments are described in detail in ref. 21. Briefly, the 169 fragments are cohesive-ended, and the 338b (b=second-generation) fragment is blunt-ended. The 169a (a = first-generation) and 169ak (k = KpnI) fragments have cohesive ends generated by PstI and KpnI restriction digestion, respectively. The cohesive ends of the 169an (n=non-equivalent termini) fragment are generated by KpnI and SacI double restriction digestion. 169 and 338b nucleosome-LH assemblies were prepared for crystallization by mixing LH and nucleosome with a slight excess of LH (1.2–1.4 molar stoichiometry) in a buffer consisting of 76–90 mM CaCl$_2$, 50 mM KCl, and 20 mM Na-acetate [pH 4.5] to give a final nucleosome-LH concentration of

4 mg ml$^{-1}$. Crystals of nucleosome-LH assembly were grown by the hanging-droplet vapor diffusion method through salting-in via equilibration against a reservoir solution containing 38–45 mM CaCl$_2$, 25 mM KCl, and 10 mM Na-acetate [pH 4.5], with incubation at 18 °C.

Nucleosome-LH assembly crystals were harvested and stabilized in a buffer consisting of 20 mM CaCl$_2$, 12.5 mM KCl, 10 mM Na-acetate [pH 4.5], 25% 2-methyl-2,4-pentanediol (MPD), and 2% trehalose. Prior to data collection, the MPD concentration was increased gradually to 65%, yielding a pronounced gain in diffraction quality. Single crystal X-ray diffraction data were recorded, subsequent to mounting stabilized crystals directly into the cryocooling N$_2$ gas stream, at beam line X06DA of the Swiss Light Source (Paul Scherrer Institute, Villigen, Switzerland) using a Pilatus 2M-F detector and an X-ray wavelength of 1.0 Å.

### Structure solution and refinement

Diffraction data were indexed, integrated, merged, scaled, and evaluated with a combination of iMosflm[45,46], XDS[47], SCALA[48], and AIMLESS[49] from the CCP4 package[50,51] and the in-house data processing pipeline, *go.com*, developed by the Swiss Light Source macromolecular crystallography beamlines (Paul Scherrer Institute, Villigen, Switzerland). Initial phases for solving the structure of the H1x-169a assembly were obtained by molecular replacement using the program PHASER[52], with the 2.2 Å resolution crystal structure of NCP composed of the 601 L DNA fragment (*pdb* code 3UT9)[53] serving as the search model. Additional search model elements for molecular replacement included the H1x NMR structure (*pdb* code 2LSO), lacking the s1-ℓ3-s2 motif. For subsequent initial structural solution of the other nucleosome-LH assemblies, search models for molecular replacement included the nucleosome from the H1x-169a structure as well as the LH$_{GD}$ from the H1x-169a structure or the H5$_{GD}$ from the assembly with a 167 bp nucleosome crystal structure (*pdb* code 4QLC)[28].

Atomic refinement and model building were carried out with REFMAC[54] and COOT[55], respectively, from the CCP4 suite[50,51]. Data collection and structure refinement statistics are given in Supplementary Tables 1 and 2.

### Structural analysis

Solvent accessible surface area calculations were conducted using GETAREA[56]. Double helix conformational parameters and reconstructions were obtained with 3DNA[57,58]. Graphic figures were prepared with PyMOL (DeLano Scientific LLC, San Carlos, CA, USA), CCPmg[59] and DSSR[60].

### Hydroxyl radical footprinting analysis

DNA or nucleosome was 5' end-labeled with $^{32}$P by mixing 12 μg of nucleosome or 6 μg of DNA, 10 units of T4 polynucleotide kinase (PNK), and 10 μCi of γ$^{32}$P-ATP (PerkinElmer, Waltham, MA, USA) in a 20 μl volume with T4 PNK buffer (70 mM Tris-HCl [pH 7.5], 1 mM MgCl$_2$, 5 mM DTT). After 1 h incubation at 37 °C, the reaction was arrested by passing the sample through a DyeEx 2.0 spin column (Qiagen, Hilden, Germany) to remove unbound γ$^{32}$P-ATP$^{32}$.P-labeled nucleosome or DNA was buffer-exchanged to 20 mM K-Cacodylate [pH 6.0] using Amicon Ultra-0.5 ml centrifugal filters (Merck Millipore, Carrigtwohill, Co. Cork, Ireland). nucleosome-LH assembly was generated by mixing H1x with nucleosome in 1.0 or 1.2 molar stoichiometry, followed by incubation on ice for 15 m.

Hydroxyl radical footprinting was carried out based on an established protocol[61]. The reaction was achieved by mixing 1 μl of 100 mM sodium ascorbate, 1 μl 0.6% H$_2$O$_2$, and 1 μl Fe$^{2+}$-EDTA (4 mM Fe$^{2+}$/8 mM EDTA) on the side of an Eppendorf tube, which was then immediately added to a 10 μl solution containing either 1 μg DNA, 2.2 μg nucleosome, or 2.4 μg nucleosome-LH assembly (in each case an equivalent quantity of DNA). After incubation for 10 m at room temperature, the reaction was arrested by adding 1 μl of 100 mM thiourea. DNA was purified by phenol:chloroform:isoamyl alcohol (25:24:1) extraction and ethanol

precipitation. After a 70% ethanol wash, the pellet was dissolved in formamide loading buffer (96% formamide, 20 mM EDTA, 0.05% (w/v) bromophenol blue). To serve as markers, a Maxam-Gilbert purine sequencing standard was prepared[62]. Samples were heated at 95 °C for 3 m before being loaded onto a denaturing 10% polyacrylamide gel, run in 0.5x TBE. Gels were dried, exposed to Imaging Screen-K (Bio-Rad Laboratories, Hercules, CA, USA), and scanned with a Typhoon Trio variable mode imager (GE Healthcare, Chicago, IL, USA).

### Kinetics and Affinity Measurements

Nucleosome-LH binding kinetics and affinity measurements were carried out using the Octet RED96e Bio-layer interferometry system (Pall Fortebio, Fremont, CA, USA). Purified 169a DNA fragment was biotinylated using enzymatic labeling, whereby biotin-14-dCTP (Thermo Scientific, Waltham, MA, USA) was coupled to the DNA 3' end by terminal deoxyribonucleotide transferase (Promega, Madison, WI, USA). Biotinylated 169a DNA was subsequently used to reconstitute a nucleosome. Biotinylated nucleosome was captured on streptavidin biosensors (Pall Fortebio, Fremont, CA, USA), pre-equilibrated with PBST buffer (8 mM Na$_2$PO$_4$, 150 mM NaCl, 2 mM KH$_2$PO$_4$, 3 mM KCl, 0.05% Tween-20 [pH 7.4]), to the extent that all sensors reached a 0.15 nm binding threshold. The captured nucleosome was stable on the surface without any drift in the baseline after loading. All binding experiments were performed at 30 °C in duplicate. LHs were determined to have no non-specific interaction with the streptavidin biosensors in PBST buffer. Single-cycle kinetics (or kinetic titration) was performed with increasing concentrations of LH (0.625, 1.25, 2.5, 5, and 10 nM). Association and dissociation phases were both measured for 300 s at each concentration. One of the biosensors was dipped in PBST buffer during both association and dissociation phases to serve as the reference channel. The data were analyzed using BIAevaluation v4.1 software (GE Healthcare, Chicago, IL, USA). The kinetic titration series obtained from the Octet was referenced and fit with a 1:1 kinetic titration model.

### Linker histone-condensate binding assays

**Purification of 601-207 × 12 template DNA.** pUC19 plasmid containing the 207 × 12 DNA template insert, composed of 12 repeats of the 601 Widom high-affinity nucleosome binding sequence[63], was purified from *E. coli* through alkaline lysis. Plasmid was digested using Xba1 and HindIII to cut out the template, and HaeII and Dra1 were used to further digest the remaining plasmid fragments (New England Biolabs, Ipswich, MA, USA). Digested plasmid fragments were separated through size exclusion chromatography on a Sephacryl S-1000 gravity-fed column. Fractions containing the 601 × 12 2500 bp fragment were combined and alcohol precipitated. The template DNA was then resuspended in 10 mM Tris-HCl (pH 7.8) and 0.1 mM EDTA buffer to yield a DNA concentration of 1 mg/mL.

**Assembly and purification of histone octamer.** Recombinant *Xenopus* core histones were purchased through the Colorado State University (CSU) Protein Expression and Purification facility. Lyophilized histones H2A, H2B, H3, and H4 were resuspended in 6 M guanidinium HCl, 20 mM Tris-HCl (pH 7.5), and 5 mM DTT for 2 hours at 25 °C. Histones were mixed at equimolar ratios to a final concentration of 1 mg/mL. The mixture was dialyzed against 2 M NaCl, Tris-HCl (pH 7.5), 1 mM EDTA, and 5 mM β-mercaptoethanol, with fresh buffer being changed three times every 4 hours[64]. Folded octamer was concentrated using Amicon Ultra-15 centrifugal filters (Merck Millipore, Carrigtwohill, Co. Cork, Ireland), and purified using size exclusion chromatography with a HiLoad 16/6- Superdex 200 column on the AKTA start liquid chromatography system (GE Healthcare, Chicago, IL, USA). Fractions collected from the peak were assessed using SDS-PAGE, then combined and concentrated to 5 mg/mL with an Amicon Ultra 15 centrifuge filter.

**Alexa 488 H4E63C labeling and labeled histone octamer assembly.** Recombinant H4E63C was purchased from the CSU Protein Expression and Purification facility. The addition of the Alexa 488 fluorophore was achieved by adding an equimolar concentration of Alexa 488 C5 maleimide (Thermo Scientific, Waltham, MA, USA) to histone H4E63C in 6 M guanidinium-HCl, 20 mM Tris-HCl (pH 7.5), and 0.7 mM tris(2-carboxyethyl)phosphine (TCEP). The fluorophore and histone solution was covered and placed at 4 °C overnight in a rotator. Fluorescently labeled H4 was then mixed with H2A, H2B, and H3C110A (to prevent additional fluorophore labeling) in equimolar ratios and subjected to the same dialysis and size exclusion chromatography purification protocol described above for unlabeled octamer assembly, with the exception that the solution was kept in the dark at all times.

**Assembly of unlabeled nucleosomal arrays and Alexa 488-labeled nucleosomal arrays.** Nucleosomal arrays were reconstituted using equimolar combinations of the purified 601 × 12 DNA template with reconstituted histone octamer at a final concentration of 0.3 mg/mL. Gradual nucleosome formation was achieved with a decreasing salt dialysis protocol at 4 °C, starting at 2 M NaCl, followed by 1 M NaCl, then 0.75 M NaCl, and ending at 2.5 mM NaCl in 10 mM Tris-HCl (pH 7.8), 0.25 mM EDTA[65]. The resulting arrays were sedimented using a Beckman XL-I analytical ultracentrifuge (Beckman Coulter, Brea, CA, USA) and analyzed using UltraScanIII software. Only arrays that sediment between 27S and 30S, which contain between 11 and 12 octamer/template[66], were used for experiments.

**Chromatin condensate formation.** Condensates were formed by adding nucleosome arrays to a 10 mM Tris-HCl (pH 7.8) and 1 mM EDTA solution containing >4 mM $MgCl_2$, to yield a final array concentration of 10 nM. This was followed by incubation for 20 minutes at 25 °C.

**Differential centrifugation pelleting assay.** Following condensate formation, nucleosome arrays were incubated with linker histone for 30 minutes at 25 °C (Supplementary Fig. 12). A given linker histone variant was mixed at a stoichiometry range encompassing 0.25, 0.5, 0.75, 1, 1.5, 2, 2.5, 3, 3.5, 4, 4.25, 4.5, 5, 5.25, 5.5, and 6 H1:nucleosome to investigate the occurrence of multiple binding modes. After incubation, condensate and linker histone mixtures were centrifuged at 20,000 g for 5 minutes, at which point condensates and condensate-bound proteins pellet while unassociated arrays and free proteins remain in the supernatant. Supernatant was removed, and the pellet was washed with a 5 mM $MgCl_2$ solution twice before the pellet and supernatant were analyzed via SDS-PAGE (Supplementary Fig. 13).

**Quantitation of condensate-bound H1.** To quantify the amount of linker histone bound to condensates, standards of known H1 quantity (μg) along the condensate pellets of various H1:nucleosome stoichiometry were analyzed by SDS-PAGE (Invitrogen Novex Wedgewell 4-20% Tris-Glycine Gels, stained with Coomassie blue; Supplementary Fig. 13). Gel images were captured using ImageQuant LAS 500 (GE Healthcare, Chicago, IL, USA). Images were analyzed via gel band densitometry using the NIH public domain Java program, ImageJ, employing the rolling ball background subtraction. Standard curves were generated for each gel based on the H1 standards and subsequently applied to determine the amount of protein in μg. At least three, and up to six, biologically independent experiments were carried out for each H1:nucleosome stoichiometry tested. In order to allow for complete and redundant coverage of all the samples, many gels were run, from which the standard slopes were calculated and applied to the condensate samples. Given the number of data points required to explore a finely sampled saturation curve, it was not possible to obtain a full range of samples and standards within a single gel.

Therefore, many gels were used to quantify condensate samples by fitting to the most proximal standard slopes.

**Fluorescence microscopy.** Alexafluor-488-H4-labeled condensate samples were incubated (covered) with varying stoichiometric ratios of unlabeled H1 variant ($r = 0$, $r = 0.5$, $r = 1.0$, $r = 2.0$, and $r = 4.0$) for 10 minutes at 25 °C. Samples were then placed on a MatTek glass bottom 35 mm petri dish with a 14 mm microwell 1.5 cover glass (MatTek, Ashland, MA, USA), covered and given an additional 20 minute incubation and settling time. Condensates were imaged using the Zeiss LSM 900 Confocal Microscope with Airyscan 2, with a 63x/1.40 objective/numerical aperture (Zeiss, White Plains, NY, USA). Zen 3.1 software was used for both capture and image analysis.

### Reporting summary
Further information on research design is available in the Nature Portfolio Reporting Summary linked to this article.

## Data availability
Atomic coordinates and structure factors for the 169a-H5 (7XVM), 169a-H1.0[L1] (6LAB), 169a-H1.0[L1b] (7XX6), 169an-H1.0 (7XVL), 169a-H1.3 (7XX5), 169a-H1x (8YTI), 338b-H1x (6L9Z), 343c-H1.0 (6LA2), 349c-H1.0 (6LA8 and 6LA9), and 353e-H1.0 (7COW) models are available at the Protein Data Bank. Additional information on the structural models analyzed here, as well as descriptions of the DNA expression constructs (available from Addgene), can be found in references [21,22]. Source data are provided with this paper. Data supporting the findings of this work are available within the article, Supplementary Information, Supplementary Data 1, and the Source Data files. Source data are provided with this paper.

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

## Acknowledgements

We are grateful to V. Olieric, M. Wang, and staff at the Swiss Light Source (Paul Scherrer Institute, Villigen, Switzerland). We thank A. Kuerzi for her contributions to the nucleosome array experiments. This project was funded by the Polish National Science Center grant DEC-2024/53/B/NZ7/03477 (C.A.D.), the Singapore Ministry of Education Academic Research Fund Tier 2/3 (MOE-T2EP30121-0005, MOE2012-T3-1-001; C.A.D.), and the National Science Foundation Grant MCB-1814012 (J.C.H.). The research leading to these results has received funding from the European Union's Horizon 2020 research and innovation program under grant agreement #730872, project CALIPSOplus (C.A.D.). The open-access publication of this article was funded by the Priority Research Area BioS under the program 'Initiative of Excellence – Research University' at the Jagiellonian University in Krakow.

## Author contributions

Investigation and methodology, all authors.; Data curation, Z.A., D.S., and C.A.D.; Analysis, validation, and visualization, Z.A., D.S., J.C.H., and C.A.D.; Conceptualization, funding acquisition, project administration, and supervision, J.C.H and C.A.D.; Writing– original draft, C.A.D.; Writing– review and editing, all authors.

## Competing interests

The authors declare no competing interests.
