## [Transparent Peer Review file · Nature Communications]

Linker Histones Consolidate Heterogenous Nucleosome Fiber Contacts by Linking Together Multiple Nucleosomes

Corresponding Author: Professor Curtis Davey

Version 0:

Reviewer comments:

Reviewer #1

(Remarks to the Author)

The manuscript by Adhireksan et al. is an extension of their earlier work published in *Nat. Commun.* 2020 & *Nucleic Acids Res.* 2021. The current study aimed to study the association of nucleosome and linker histone subtypes (Human H1x, H1.0, H1.3 and Chicken H5) in oligo-nucleosome chromatin fibers at an atomic level using X-ray crystallography. The authors found that LHs bind in 'non-canonical' off-dyad modes, associating with the inter-nucleosomal linker DNA and other positions on the nucleosome. They find that in many cases LHs engage in multiple bridging interactions between DNA segments nucleosomes, with some variation between the LH subtypes studied. The study reports the first structural visualization of alternative H1 binding modes (beyond the canonical on-dyad mode) in a chromatin complex, extending our knowledge of how LH subtypes interact in non-canonical ways in chromatin, especially in situations where there is stoichiometric excess of LH over nucleosomes. However, enthusiasm is somewhat damped by the low resolution for two of the four structures, some issues with the way the data is analyzed and presented, some lack of connection to prior work, and discussion of limits of the current study.

Comments:

In the abstract and other places the authors state that the commonly accepted mode of LH binding is based on studies of individual nucleosomes. However, the paper ignores studies of H1-di- and tri-nucleosome complexes (Syed et al., 2010 PNAS) and the crystal structure of a 6-nucleosome array bound to histone H1 (Garcia-Saez, 2018 Mol. Cell), both showing on-dyad binding to oligonucleosomes is identical to full-length H1 binding to mononucleosomes.

Relatedly, the 'closed-loop' dinucleosomes structures appear to force a full cross-over of linker DNA, perhaps interfering with some LH binding to the canonical on-dyad site. Is this the case, i.e. are there steric hinderances or distortions that would inhibit LHG domain fitting in at the canonical on-dyad position? This should be discussed/presented in the MS.

Results: The authors show an 'on-dyad' mode of binding for H5, but there is no comparison to previous reports of on-dyad binding modes for this or other LHs (Zhou et al., 2015 Mol. Cell; Bednar et al., 2017 Mol. Cell; Zhou et al., 2021 Mol. Cell).

Page 7, Authors statement that "compaction of Ns within the crystal is fostered by DNA binding of LH, N- and C-terminal tails of the core histones and Ca²⁺ ions, which function in large part to provide electrostatic counterparts for the double helix". The authors should show or document a comparison of structures in absence and presence of LH to support this conclusion.

Page 7, Author's mentioned that based on crystallization outcome LHs present between the linker DNA or within the roughly cube shaped cavity. Please label this cavity in Fig. 1. In addition, it would be helpful to the reader to see an animation/video of the different LHs positioning between the linker DNAs and in the cavity to support Fig. 2 (besides Ref. 20&21), and furthermore tabulate the distances/interactions between LHs and nucleosomes/DNA for Fig. 3, 4, and 5b to provide better understanding of the interacting residues at an atomic level beside their surface accessibility as mentioned in Table S2.

Figure S3, Some of the electron density for LHNT and LHCT has been found and the region has been modelled and marked

to interact with the distal nucleosome. It would be interesting to show a pictorial view of the interaction and tabulate the distances of the interacting residues as well. Similarly, indicate the LHNT interactions for Figure S11.

Figure 8 should be extended by showing individual orientation with interacting residues followed by a combined model, it's always nice to show the interacting partners/resides in tabulated form.

The authors report extremely tight binding for H1-N interactions. Given that the authors detail multiple alternative binding modes for these LHs, how well does the 1:1 kinetic binding model fit the data? If >1 H1 binding sites exist per nucleosome, how does this affect the calculation of overall affinity?

Discussion: Is there potential for numerous additional alternative H1-binding modes in chromatin beyond what is reported here?

Minor comments:

Results, p. 5 bottom: The authors state: "Various constructs were also co-crystallized with different LH variants, which included human H1x, H1.0, H1.1–H1.5 and chicken H5 (a.k.a. avian H1.0)". I found this presentation to be confusing when no results are presented for most of these proteins. Simply say that 'similar attempts were made with LHs X, Y, and Z, but insufficient density was detected to allow analysis for these proteins'?

Fig. 1A. What does the cloud around the LH cartoon structures indicate? This should be noted in the figure legend.

Fig. 2. The current representation of Fig. 2a-b is somewhat crowded and its hard to the layering. Again, a video or animation would help the reader here.

Similarly, the authors state: "With one particular type of construct, which consists of a 165 bp double helix having 4-nucleotide overhangs at either terminus, we were able to solve many structures with the four different LH variants." Again, confusing. What is meant by "many structures"? Is the linker DNA evenly distributed about the nucleosome core binding site? Shouldn't "either terminus" be "both termini"?

Results, p. 6, top: Related, the authors state "However, the best diffracting crystals were typically the ones co-crystallized with H1.0 or H1x, and these two variants together with H5, and to a lesser extent H1.3, were the only LH types for which the associated electron density was sufficiently clear to permit explicit modeling of at least the LHGD (Table S1 & Figs. S1–S8). This is a long and confusing sentence.

Related, p. 7, middle para: "A total of 11 LHGD of H1x, H1.0, H1.3 and H5 across 7 different N-LH structures, encompassing nearly identical N packing (lattice) configurations, all adopt the winged helix fold and are overall similar to each other (Fig. 2c). Again, long, cumbersome, confusing to the reader.

What are the DNA sequences for the templates, and specifically what do a, ak, and an mean in terms of sequence?

Fig. S1-S8: Choose contrast color and bolding for numeric labelling of residues for better visualization.

Page 7, in my opinion the statement "Electron density maps stemming from H1x-169a assembly crystals are especially well defined" should be specified:well defined H1x globular domain.

Figure S9, please mark the interface with a dotted line and labelled the respective nucleosomes as shown in S9a.

Please expand what the author means in the header of Table S1. I understand it makes more sense to show the data set of H1x-169aP, H1.0-169aN1, H1.0-169anR, H1.3-169aF, H5-169aD/N2 but what about the data set with similar nucleosome conditions (mono or di)?

Reviewer #2

(Remarks to the Author)

The manuscript by Adhireksen et al. builds on previous work by the authors reported in ref. 18, in which they engineered a number of nucleosomal constructs with cohesive-ended DNA that allowed them to crystallize nucleosomes in a circular or continuous fibre configuration. The current manuscript reports five additional such structures at resolutions ranging from 2.7 to 3.5 Å. These describe nucleosomes in complex with four different linker histone (LHs) - the human histones H1.0 (two structures), H1x and H1.3 as well as chicken histone H5. The new structures all comprise a circular configuration of a nucleosome dimer, in which the entry and exit linker arms of one nucleosome connect to the exit and entry arms of the other, respectively. These structures show similar crystal packing and contain the same palindromic DNA sequence (169a), except for an H1.0 structure that contains a non-palindromic sequence (169an). The crystal structures capture the LHs in three previously observed configurations (on-dyad, off-dyad and peripheral) as well as three new configurations that the authors term reverse peripheral, non-dyad(1) and non-dyad(2).

The authors also report complementary data that include: bio-layer interferometry data showing that the four LHs bind the 169a mononucleosome with similar affinity; a hydroxyl radical footprint showing that H1x binds to the dyad and linker DNA of the 169a nucleosome; fluorescence microscopy data showing that the addition of H1x or H1.0 alters the appearance of nucleosomal condensates; and binding data indicating that H1x and H1.0 can associate with condensates in a molar ratio exceeding 1 LH per nucleosome.

Overall, the reported structures represent an impressive technical achievement, with the new observed LH configurations providing novelty. However, to what extent the nucleosomal packing in these engineered crystal forms is relevant to the cellular chromatin context is unclear, as is the physiological relevance of the novel observed LH configurations. Also, the authors observe certain LH configurations associated with specific H1 variants. However, whether this H1 variant dependence sheds light on the functional diversity of H1 variants in the cell or is merely a consequence of the specific crystal packing constraints associated with the engineered nucleosome constructs is an open question. The link between the different reported LH configurations and the condensate binding data reported in Figure 7 is also somewhat tenuous.

Most remarkably, the authors make the claim that they could trace the entire N- and C-terminal domains of H1.x. If true, this would be a highly significant and original result. Unfortunately, 2Fo-Fc and omit maps calculated from the data deposited by the authors (PDB 7xx7) reveal that the claim is unfounded. The density for residues 1-35 and 120-213 is uninterpretable. The atomic model for these regions contains long stretches of residues devoid of any secondary structure or of any contacts with neighbouring molecules, rendering a unique stable conformation highly improbable. The geometry for these regions is also very poor. In short, the atomic model proposed for the H1.x N- and C-terminal domains appears inconsistent with the diffraction data and with stereochemical considerations.

Specific points.

1. On p.7 the authors state: *"A total of 11 LHGD copies of H1x, H1.0, H1.3 and H5 across 7 different N-LH structures, encompassing nearly identical N packing (lattice) configurations, all adopt the winged helix fold and are overall similar to each other (Fig. 2c)."* To which 11 LHGD copies and 7 N-LH structures are the authors referring? Table S1 lists only 5 PDB structures and inspection of these reveals a total of only 8 LH polypeptide chains.
2. Related to the previous point, Fig. 2c shows H1.0 as adopting three configurations: non-dyad, peripheral and reverse-peripheral. However, the two corresponding PDB structures (7xx6 and 7xvl) only show the non-dyad configuration (occurring twice) and the reverse peripheral configuration. Which structure shows H1.0 as adopting a peripheral configuration? (Is it 6lab from ref.18 ? If so, this should be indicated.)
3. On p.10, the authors state *"Nonetheless, the $\alpha 1/\alpha 3$ motif is in fact employed in a distinct fashion, whereby the association is with an SHL 4 minor groove and coincides with remodeling of the H1.0GD conformation to accommodate the distinct binding context of this mode (Figs. 3, 4 & 5a). Specifically, the register of $\alpha 3$ is rotated by ~ 1 residue ($\sim 100^\circ$) relative to the rest of the GD as compared to the other structures for H1.0. This indicates a degree of plasticity inherent in the H1.0GD that would help foster the variety of different modes displayed by this variant."*
The notion that the register of helix $\alpha 3$ for this structure (PDB 7xx6) is shifted by 1 residue relative to other H1.0 structures is quite an extraordinary claim that would require strong justification. In actual fact, helix $\alpha 3$ for PDB 7xx6 is in perfect register with helix $\alpha 3$ of a previously reported human H1.0 structure (PDB 7k5x) and with the available AlphaFold model. Rather, it is helix $\alpha 3$ of the authors' H1.0-169an structure (PDB 7xvl) that is out of register with these other models. Of the 5 structures reported in this manuscript, 7xvl has the lowest resolution (3.5 Å). The authors appear to have introduced an error while refining this structure, since the quality and low resolution of the electron density does not justify rotating the helix by 100° .
4. Section entitled "Tails Provide Electrostatic Glue", p.15. Given the comments above regarding the inappropriate modeling of the N- and C-terminal domains of H1.x, most of the statements made in this section are completely untethered to any observational data.
5. Regarding the binding data shown in Fig. S16:
 - (i) Why does panel (d) only show the molar input ratios 5.25-6 for H1x and not 1-6 as in panel (c) ?
 - (ii) The standards shown on the right-hand side of panel (c) cover only the low end of the range of band intensities observed on the left-hand part of the gel. Was the standard curve used in Table S4 generated from only this limited set of data ? (Notably, the condensate binding curve in Figure 7a shows a conspicuous jump between $rH1 = 3.5$ and 4 for H1.0. Might this reflect a quantitation error based on an extrapolated standard curve?)
6. Many of the structural figures are somewhat difficult to interpret. In certain cases, it was necessary to view the atomic coordinates of the deposited PDB file in order to better understand the figure. A small inset showing a global view of the dinucleosome might help orient the reader.

Version 1:

Reviewer comments:

Reviewer #1

(Remarks to the Author)

The authors have adequately addressed all of our queries and comments in the revised MS. However I would suggest the authors include a table of all 10 constructs that yielded well-diffracting crystals to complement the initial part of the Results, providing general information about each. This would help readers keep straight the PDB identifiers for each, which contained H1s, what H1 they contained, which H1s had sufficient densities for modeling and which did not, what H1 binding modes were detected in each structure, and overall resolutions for each of the 10. It would be helpful to also list the 14 constructs that did not yield well-diffracting crystals, perhaps in the table legend.

I will also note that H1s forming general bridging interactions between multiple DNA segments was first elucidated by Jean Thomas and David Clark (cf JMB 1986 187:569; EJB 1988 178:225). Given the multiple DNA bridging modes by H1s described in the current MS, it seems appropriate to cite this work.

Reviewer #2

(Remarks to the Author)

My comments are in the accompanying PDF file.

Version 2:

Reviewer comments:

Reviewer #2

(Remarks to the Author)

My evaluation of the initial version of this manuscript submitted in 2023 raised two broad categories of concerns: (i) issues regarding the validity of the crystallographic models and the scientific claims derived from them, and (ii) issues concerning the biological relevance of the linker-histone binding modes observed in the authors' engineered crystal systems. As the concerns in category (i) were not adequately addressed in the authors' response letter, my subsequent evaluation focused exclusively on these deficiencies and deferred any assessment of how they had addressed the concerns in category (ii). The authors have now satisfactorily resolved the first category: the revised structural models have been corrected for the major errors identified previously, and the crystallographic foundations of the study are substantially improved. However, the second category of concerns remains unresolved, as the manuscript still does not establish that the structural observations made in this highly engineered system have meaningful implications for how linker histones function *in vivo*.

The central finding of this study is that H1 globular domains can bridge distinct DNA segments through various canonical and non-canonical modes. While the authors' analyses of on- and off-dyad binding modes add only incrementally to earlier structural studies from several groups, their detailed examination of the linking modes (L1, L1b, L2 and L2b) extends and elaborates on the non-dyad (L2) binding mode they previously reported using a related construct (Adhireksan et al., Nat Commun 2020), thereby expanding the catalogue of H1-DNA interaction modes observed *in vitro*. Thus, there is some structural novelty here, although the conceptual advance that a non-dyad binding mode is possible was already established in their 2020 study.

Importantly, however, these findings do not demonstrate that the linking modes observed in the crystal structures are used, or are even accessible, in physiological chromatin. The data and arguments presented in support of biological relevance are not sufficiently persuasive. In particular, the observation that condensates can bind linker histones at stoichiometries above 1 is of uncertain biological significance. (The authors point to the existence of cell types with H1:nucleosome ratios above 1 as supporting evidence, but the cited reference [Woodcock et al., 2006] reports H1:nucleosome ratios below 1 in most systems, with values slightly above 1 observed only in highly specialized contexts). More importantly, the array experiments do not show that H1 within the condensates adopts the L1 or L2 binding modes reported in the crystal structures, raising the possibility that these modes may be *in vitro* phenomena specific to their crystallization system.

In conclusion, while the structural analyses have some interest, the study does not appreciably advance our understanding of linker histone function *in vivo*, which limits its overall impact for the chromatin field.

Reviewer #3

(Remarks to the Author)

The issues are:

(1) Do the new structural models represent an important contribution to the chromatin literature? There were a surprising number of errors in the structures described in the first version of this paper. However, the structures have been corrected. Although they are variants on the authors' previous interdigitated fibre model (Adhireksan et al. ref. 22), it is important to understand the various binding modes of H1 variants within nucleosomal fibres. This paper describes differences between binding modes in structures containing H1x, H10 or H5 and is therefore an important addition to the literature.

(2) Do the various linker histone binding modes observed have any biological significance? The large majority of the literature supports a stoichiometry of up to one H1 molecule per nucleosome. Consequently, chromatin structures with more than one H1 per nucleosome, such as those presented in this paper, might not be biologically important. However, there is one clear and relevant example of chromatin with more than one H1 molecule per nucleosome: transcriptionally inert chicken erythrocyte chromatin has ~0.9 molecules of H5 and ~0.4 molecules of H1 (Bates & Thomas, 1981; PMID 7312631). In that paper, the stoichiometry of linker histone to core histones was also measured very carefully for lymphocyte and glial chromatin, finding one H1 per nucleosome in both cases. This paper could be cited as support for the biological relevance of models with more than one linker histone per nucleosome (at least for those including H5). In their Discussion, the authors note that even if the average stoichiometry is one linker histone per nucleosome, H1 molecules might not be distributed evenly in chromatin, resulting in some regions with two linker histones per nucleosome and others with none. This

suggestion is supported by refs. 34 and 35.

(3) The use of 'N' for 'nucleosome' does not seem necessary.

RESPONSES TO REVIEWER COMMENTS

We thank the reviewers wholeheartedly for their valuable time and input on our manuscript. We have carefully considered the recommendations of the reviewers, whose insightful comments have been very helpful in compiling the revised version. Below we outline the revisions implemented for each of the points raised by the reviewers.

In response to comments by Reviewer #1:

1. *The manuscript by Adhireksan et al. is an extension of their earlier work published in Nat. Commun. 2020 & Nucleic Acids Res. 2021. The current study aimed to study the association of nucleosome and linker histone subtypes (Human H1x, H1.0, H1.3 and Chicken H5) in oligo-nucleosome chromatin fibers at an atomic level using X-ray crystallography. The authors found that LHs bind in ‘non-canonical’ off-dyad modes, associating with the inter-nucleosomal linker DNA and other positions on the nucleosome. They find that in many cases LHs engage in multiple bridging interactions between DNA segments nucleosomes, with some variation between the LH subtypes studied. The study reports the first structural visualization of alternative H1 binding modes (beyond the canonical on-dyad mode) in a chromatin complex, extending our knowledge of how LH subtypes interact in non-canonical ways in chromatin, especially in situations where there is stoichiometric excess of LH over nucleosomes. However, enthusiasm is somewhat damped by the low resolution for two of the four structures, some issues with the way the data is analyzed and presented, some lack of connection to prior work, and discussion of limits of the current study.*

Response: We appreciate the reviewer’s comments. We have addressed the general criticisms raised here by responding to the specific comments outlined below.

Comments:

2. *In the abstract and other places the authors state that the commonly accepted mode of LH binding is based on studies of individual nucleosomes. However, the paper ignores studies of H1-di- and tri-nucleosome complexes (Syed et al., 2010 PNAS) and the crystal structure of a 6-nucleosome array bound to histone H1 (Garcia-Saez, 2018 Mol. Cell), both showing on-dyad binding to oligonucleosomes is identical to full-length H1 binding to mononucleosomes.*

Response: We have modified the abstract and incorporated these studies into the description within the Introduction section (currently, references 16 and 17).

3. *Relatedly, the ‘closed-loop’ dinucleosomes structures appear to force a full cross-over of linker DNA, perhaps interfering with some LH binding to the canonical on-dyad site. Is this the case, i.e. are there steric hinderances or distortions that would inhibit LHG domain fitting in at the canonical on-dyad position? This should be discussed/presented in the MS.*

Response: We have incorporated a discussion of this issue at the end of the *Linker Histones Bind Multiple Nucleosomes in Diverse Modes* section: “The ability of H5 to associate with the 169a N dimer via the on-dyad mode notably indicates that this binding site is available to the LH_{GD} in the 169 crystals, in spite of structural constraints that may be imposed by the nature of N pairing. Nonetheless, the absence of on-dyad H5 binding in the second, opposing, N of the dimer may be a consequence of either a special favorability associated with the observed H5 occupancy at the non-dyad binding site or an on-dyad-disfavoring/incompatible N conformation (in any case, a significant degree of plasticity is inherent in the nature of on-dyad LH_{GD}-N interactions; *vide infra*).”

4. Results: *The authors show an ‘on-dyad’ mode of binding for H5, but there is no comparison to previous reports of on-dyad binding modes for this or other LHs (Zhou et al., 2015 Mol. Cell; Bednar et al., 2017 Mol. Cell; Zhou et al., 2021 Mol. Cell).*

Response: We have included a discussion of these works together with a structural comparison (Supplementary Fig. 18) in the first paragraph of the Discussion section.

5. Page 7, Authors statement that “compaction of Ns within the crystal is fostered by DNA binding of LH, N- and C-terminal tails of the core histones and Ca²⁺ ions, which function in large part to provide electrostatic counterparts for the double helix”. The authors should show or document a comparison of structures in absence and presence of LH to support this conclusion.

Response: We have further supported this statement by including (in the *Nucleosome-Linker Histone Assembly Structures* section) a comparison of the counterbalancing charges in the structure, also with reference to previous works (refs. 24 & 25) on LH-free nucleosome structure.

6. Page 7, Author’s mentioned that based on crystallization outcome LHs present between the linker DNA or within the roughly cube shaped cavity. Please label this cavity in Fig. 1. In addition, it would be helpful to the reader to see an animation/video of the different LHs positioning between the linker DNAs and in the cavity to support Fig. 2 (besides Ref. 20&21), and furthermore tabulate the distances/interactions between LHs and nucleosomes/DNA for Fig. 3, 4, and 5b to provide better understanding of the interacting residues at an atomic level beside their surface accessibility as mentioned in Table S2.

Response: The cube-shaped cavity has been indicated in Fig. 1b. We have included an animation to accompany Figure 2b (Supplementary Movie 1). In addition, we have included a tabulation, as Supplementary Data 1, of the LH-nucleosome H-bonding interactions to support the figures.

7. Figure S3, Some of the electron density for LHNT and LHCT has been found and the region has been modelled and marked to interact with the distal nucleosome. It would be interesting to show a pictorial view of the interaction and tabulate the distances of the interacting residues as well. Similarly, indicate the LHNT interactions for Figure S11.

Figure 8 should be extended by showing individual orientation with interacting residues followed by a combined model, it’s always nice to show the interacting partners/resides in tabulated form.

Response: A panel (c) has been added to Supplementary Figure 3 to illustrate the 3D H1.0 tail structure and nucleosome interactions, which are also tabulated in Supplementary Data 1. Figure 9 (formerly Figure 8) has been extended by showing the five structures individually together with interacting residues (in stereo view); panels a through e of Supplementary Figure 17.

8. The authors report extremely tight binding for H1-N interactions. Given that the authors detail multiple alternative binding modes for these LHs, how well does the 1:1 kinetic binding model fit the data? If >1 H1 binding sites exist per nucleosome, how does this affect the calculation of overall affinity?

Response: The 1:1 LH:N binding model resulted in a conclusive fit to the sensorgram profiles for both of the replicates (Supplementary Fig. 14). In any case, the density of the surface-immobilized nucleosomes is sufficiently sparse so as not to permit interaction of one LH molecule with more than one nucleosome. The opposing scenario, whereby a single nucleosome interacts with multiple LH molecules is neither expected nor observed. For

instance, in the native-PAGE analysis of Supplementary Fig. 10, LH titration to the 169a nucleosome (the same used in the kinetic/BLI measurements) yields only 1:1 complex up to higher stoichiometry, at which point large aggregates form. The 15–45 pM range of K_D values found for the LH-169a system is in close agreement with solution-state fluorescence assay (HI-FI FRET) measurements on the binding of H1 to a variety of nucleosomal constructs (White et al., 2016; PMID: 26750377).

9. *Discussion: Is there potential for numerous additional alternative H1-binding modes in chromatin beyond what is reported here?*

Response: We have added a comment in this regard in the second paragraph of the Discussion section: “In fact, the heterogenous nature of local N organization observed in cellular chromatin^{30,31} might suggest the existence of many distinct linker histonophores, and thus ‘non-dyad’ modes (that is, potentially well beyond what we have observed so far).”

Minor comments:

10. *Results, p. 5 bottom: The authors state: “Various constructs were also co-crystallized with different LH variants, which included human H1x, H1.0, H1.1–H1.5 and chicken H5 (a.k.a. avian H1.0)”. I found this presentation to be confusing when no results are presented for most of these proteins. Simply say that ‘similar attempts were made with LHs X, Y, and Z, but insufficient density was detected to allow analysis for these proteins’?*

Response: This sentence and the corresponding first paragraph of the Results section have been reworded.

11. *Fig. 1A. What does the cloud around the LH cartoon structures indicate? This should be noted in the figure legend.*

Response: The Figure 1 legend has been amended to include a description of this feature (which was added to emphasize the LHs is this first depiction).

12. *Fig. 2. The current representation of Fig. 2a-b is somewhat crowded and its hard to the layering. Again, a video or animation would help the reader here.*

Response: We have included an animation to accompany Figure 2b (Supplementary Movie 1).

13. *Similarly, the authors state: “With one particular type of construct, which consists of a 165 bp double helix having 4-nucleotide overhangs at either terminus, we were able to solve many structures with the four different LH variants.” Again, confusing. What is meant by “many structures”? Is the linker DNA evenly distributed about the nucleosome core binding site? Shouldn’t “either terminus” be “both termini”?*

Response: This description (in the second paragraph of the Results section) has been reworded.

14. *Results, p. 6, top: Related, the authors state “However, the best diffracting crystals were typically the ones co-crystallized with H1.0 or H1x, and these two variants together with H5, and to a lesser extent H1.3, were the only LH types for which the associated electron density was sufficiently clear to permit explicit modeling of at least the LHGD (Table S1 & Figs. S1–S8). This is a long and confusing sentence.*

Response: This sentence and the corresponding first paragraph of the Results section have been reworded.

15. *Related, p. 7, middle para: “A total of 11 LHGD of H1x, H1.0, H1.3 and H5 across 7 different N-LH structures, encompassing nearly identical N packing (lattice) configurations, all adopt the winged helix fold and are overall similar to each other (Fig. 2c). Again, long, cumbersome, confusing to the reader.*

Response: This sentence has been split and reworded.

16. *What are the DNA sequences for the templates, and specifically what do a, ak, and an mean in terms of sequence?*

Response: We have added a brief description of these DNA fragments in the first paragraph of the Methods section. Further details are available in Adhireksan et al., 2021 (ref. 20).

17. *Fig. S1-S8: Choose contrast color and bolding for numeric labelling of residues for better visualization.*

Response: Supplementary Figures 1 through 8 have all been revised so that the residue numbers stand out.

18. *Page 7, in my opinion the statement “Electron density maps stemming from H1x-169a assembly crystals are especially well defined” should be specified:well defined H1x globular domain.*

Response: We appreciate the reviewer’s point here, but our intension in this opening statement is to make an overall qualification of this data set (or crystal type). The electron density clarity allowed us to model other features as well, including solvent species (Ca²⁺ ions, etc.) and core histone tails, which was generally not feasible with the other data sets. In any case, the quality of the electron density associated with the globular versus the tail domains is described/presented in detail later in the manuscript.

19. *Figure S9, please mark the interface with a dotted line and labelled the respective nucleosomes as shown in S9a.*

Response: We have demarcated the nucleosome-nucleosome interface in Supplementary Figure 9a and signified the distinct nucleosomal components in Supplementary Figures 9b and 9c.

20. *Please expand what the author means in the header of Table S1. I understand it makes more sense to show the data set of H1x-169aP, H1.0-169aN1, H1.0-169aN2, H1.3-169aF, H5-169aD/N2 but what about the data set with similar nucleosome conditions (mono or di)?*

Response: To Supplementary Table 1, we have added a note, which explains that the H1.0^P-169a and H1x^P-338b data/refinement were described earlier in Adhireksan *et al.*, 2021 (ref. 20).

In response to comments by Reviewer #2:

21. *The manuscript by Adhireksen et al. builds on previous work by the authors reported in ref. 18, in which they engineered a number of nucleosomal constructs with cohesive-ended DNA that allowed them to crystallize nucleosomes in a circular or continuous fibre configuration. The current manuscript reports five additional such structures at resolutions ranging from 2.7 to 3.5 Å. These describe nucleosomes in complex with four different linker histone (LHs) - the human histones H1.0 (two structures), H1x and H1.3 as well as chicken*

histone H5. The new structures all comprise a circular configuration of a nucleosome dimer, in which the entry and exit linker arms of one nucleosome connect to the exit and entry arms of the other, respectively. These structures show similar crystal packing and contain the same palindromic DNA sequence (169a), except for an H1.0 structure that contains a non-palindromic sequence (169an). The crystal structures capture the LHs in three previously observed configurations (on-dyad, off-dyad and peripheral) as well as three new configurations that the authors term reverse peripheral, non-dyad(1) and non-dyad(2).

The authors also report complementary data that include: bio-layer interferometry data showing that the four LHs bind the 169a mononucleosome with similar affinity; a hydroxyl radical footprint showing that H1x binds to the dyad and linker DNA of the 169a nucleosome; fluorescence microscopy data showing that the addition of H1x or H1.0 alters the appearance of nucleosomal condensates; and binding data indicating that H1x and H1.0 can associate with condensates in a molar ratio exceeding 1 LH per nucleosome.

Overall, the reported structures represent an impressive technical achievement, with the new observed LH configurations providing novelty. However, to what extent the nucleosomal packing in these engineered crystal forms is relevant to the cellular chromatin context is unclear, as is the physiological relevance of the novel observed LH configurations. Also, the authors observe certain LH configurations associated with specific H1 variants. However, whether this H1 variant dependence sheds light on the functional diversity of H1 variants in the cell or is merely a consequence of the specific crystal packing constraints associated with the engineered nucleosome constructs is an open question. The link between the different reported LH configurations and the condensate binding data reported in Figure 7 is also somewhat tenuous.

Response: We appreciate the reviewer's comments. Regarding the H1 variant dependence, our inferences are also supported by the fact that the 169a, 169an, and 338b crystals, which encompass the H1x, H1.0, H1.3, and H5 structures, are either nearly isomorphous, or at least have very similar overall nucleosome packing. Therefore, it can be assumed that differences observed between the LH variants within this set of crystals, grown and stabilized under identical conditions, are a reflection of distinct nucleosome binding predispositions. We also note that in most instances, we had collected and inspected multiple X-ray data sets for a given compound/crystal type, thus screening for the existence of potential crystal polyforms that could display distinct LH binding. Different crystals/batches with H1x, H5, and H1.3 all display the same binding mode(s) within the LH variant set, whereas for H1.0, different modes are seen between crystals of even the exact same composition. Moreover, since H5 is reproducibly seen to associate on-dyad (in the one copy per nucleosome dimer), while other variants are not, this also speaks for a variant-dependent predisposition with respect to this particular 'canonical' binding mode. In fact, the LH-condensate binding data also reveals variant-dependent behavior.

With respect to the link between the crystal structure and condensate binding data, one could envision an alternative scenario, in which LH titration of the condensates would show binding saturation around 1:1 H1:nucleosome stoichiometry and moreover not show linking together/aggregation of the individual condensates. In this hypothetical situation, we would not be able to argue that there must be alternative/additional binding modes, since a valid assumption could be that the single binding site occupied per nucleosome is the on-dyad mode (or a mix of on- and off-dyad modes). Furthermore, based on that hypothetical data, we could also not argue that H1 is active in linking multiple nucleosomes, that is, multiple sections of nucleosome fiber, together. Therefore, we argue that the condensate data decisively strengthens the crystallographic/structural observations.

22. *Most remarkably, the authors make the claim that they could trace the entire N- and C-terminal domains of H1.x. If true, this would be a highly significant and original result. Unfortunately, 2Fo-Fc and omit maps calculated from the data deposited by the authors (PDB 7xx7) reveal that the claim is unfounded. The density for residues 1-35 and 120-213 is uninterpretable poor. The atomic model for these regions contains long stretches of residues devoid of any secondary structure or of any contacts with neighbouring molecules, rendering a unique stable conformation highly improbable. The geometry for these regions is also very poor. In short, the atomic model proposed for the H1.x N- and C-terminal domains appears inconsistent with the diffraction data and with stereochemical considerations.*

Response: We do not agree with the premise that the H1x N- and C-terminal tail structures are not supported by the data. We went to great lengths to carefully model the tail structures and also to document the support of the modeled conformations with omit electron density maps (Supplementary Figures 2 and 3). As we noted above and in the manuscript, the electron density quality over the tail regions fluctuates significantly, but there is information content (supported by the omit maps), which importantly illustrates the character of the H1 tails. In any case, however, we want to make it clear that there are apparently alternative tail conformations sampled in the H1x-169a crystals and most likely multiple, if not many, distinct conformations available to the tails in other nucleosomal contexts (i.e., in chromatin). As such, we have added a further discussion of the tails following the second paragraph of the Discussion section.

By analogy to the core histone tails, the H1x tails conformational behavior is consistent with observations that the N- and C-terminal linker histone tails are unstructured in solution (reviewed in McBryant and Hansen, 2012, for instance). In fact, in the H1x-169a structure, the linker histone tails display structural and electron density characteristics that are fully analogous to that seen previously with the core histone tails at high and lower resolutions (i.e., ~2–3 Å; for example, Davey et al., 2002), whereby the tails undergo multiple disorder-to-order transitions as they ‘meander’ through the lattice. In particular, the most ordered regions of the tails coincide with contacts to other protein or DNA elements. For the tail regions that are the most disordered, there are likely to be alternative (multiple) conformations that connect to the better ordered, or interacting, regions. Although the H1x tail regions are almost completely devoid of regular secondary structural features, this would be consistent with their richness in and regular intervals of proline and glycine residues.

Specific points.

23. *1. On p.7 the authors state: "A total of 11 LHGD copies of H1x, H1.0, H1.3 and H5 across 7 different N-LH structures, encompassing nearly identical N packing (lattice) configurations, all adopt the winged helix fold and are overall similar to each other (Fig. 2c)." To which 11 LHGD copies and 7 N-LH structures are the authors referring? Table S1 lists only 5 PDB structures and inspection of these reveals a total of only 8 LH polypeptide chains.*

Response: The legend for Figure 2 has been amended to specify the structures in this analysis: “(c) Superimposition of the LH_{GD} regions from the 169a (pdb codes 7XX7 [H1x]₂, 6LAB [H1.0]₂ [ref. 20], 7XX6 [H1.0]₂, 7XX5 [H1.3], 7XVM [H5]₂), 169an (pdb code 7XVL [H1.0]), and 338b (pdb code 6L9Z [H1x] [ref. 20]) crystal structures (11 LH molecules in total; colored by variant type).” The text has been further amended elsewhere to specify the respective pdb codes involved in the analyses.

24. *2. Related to the previous point, Fig. 2c shows H1.0 as adopting three configurations: non-dyad, peripheral and reverse-peripheral. However, the two corresponding PDB structures (7xx6 and 7xvl) only show the non-dyad configuration (occurring twice) and the reverse*

peripheral configuration. Which structure shows H1.0 as adopting a peripheral configuration? (Is it 6lab from ref.18 ? If so, this should be indicated.)

Response: Yes, the H1.0-169a structure coinciding with the peripheral mode is 6LAB, which was originally disclosed in Adhireksan et al., 2021 (therein Table 2, “169a”). The text has been amended to reflect this and otherwise elsewhere to specify the respective pdb codes involved in the analyses.

25. 3. *On p.10, the authors state "Nonetheless, the $\alpha 1/\alpha 3$ motif is in fact employed in a distinct fashion, whereby the association is with an SHL 4 minor groove and coincides with remodeling of the H1.0_{GD} conformation to accommodate the distinct binding context of this mode (Figs. 3, 4 & 5a). Specifically, the register of $\alpha 3$ is rotated by ~1 residue (~100°) relative to the rest of the GD as compared to the other structures for H1.0. This indicates a degree of plasticity inherent in the H1.0_{GD} that would help foster the variety of different modes displayed by this variant." The notion that the register of helix $\alpha 3$ for this structure (PDB 7xx6) is shifted by 1 residue relative to other H1.0 structures is quite an extraordinary claim that would require strong justification. In actual fact, helix $\alpha 3$ for PDB 7xx6 is in perfect register with helix $\alpha 3$ of a previously reported human H1.0 structure (PDB 7k5x) and with the available AlphaFold model. Rather, it is helix $\alpha 3$ of the authors' H1.0-169an structure (PDB 7xvl) that is out of register with these other models. Of the 5 structures reported in this manuscript, 7xvl has the lowest resolution (3.5 Å). The authors appear to have introduced an error while refining this structure, since the quality and low resolution of the electron density does not justify rotating the helix by 100°.*

Response: We thank the reviewer for pointing this out, and while it is not an error in our modeling, we have added an analysis/discussion of this issue in the *Linker Histones Bind Multiple Nucleosomes in Diverse Modes* section: "...[regarding the H1.0-169a non-dyad polymorph,] Specifically, the register of $\alpha 3$ is rotated by ~1 residue (~100°) relative to the rest of the GD as compared to that in the other H1.0-169a structure (showing the peripheral mode) and the H1.0-169an structure (coinciding with the reverse-peripheral mode). However, the two peripheral-type structures are in fact the exceptions, as the $\alpha 3$ register seen in this H1.0-169a non-dyad polymorph is apparently the common, 'consensus/canonical', H1.0_{GD} helical register that is seen in all of our other H1.0 on-dyad/non-dyad structures^{20,21} (Fig. 4: H1.0^{D*}, H1.0^{N*}, H1.0^{N1}; Fig. 9) in addition to that also observed in a recent cryo-EM structure of the H1.0-chromatosome²⁶. In any case, the structural polymorphism indicates a degree of plasticity inherent in the H1.0_{GD} that would help foster the variety of different modes displayed by this variant." In addition, we also note that the peripheral-mode H1.0-169a structure, at 3.2 Å resolution (originally disclosed in Adhireksan et al., 2021), displays the same register as the 3.5 Å-resolution reverse peripheral-mode H1.0-169an structure. The omit electron density maps support the modeled configurations, and the distinctions in register of the $\alpha 3$ helix also coincide with concerted changes in neighboring GD elements that help rationalize the overall conformations that are adopted.

26. 4. *Section entitled "Tails Provide Electrostatic Glue", p.15. Given the comments above regarding the inappropriate modeling of the N- and C-terminal domains of H1.x, most of the statements made in this section are completely untethered to any observational data.*

Response: We do not agree with the premise that the H1x N- and C-terminal tail structures are not supported by the data. We went to great lengths to carefully model the tail structures and also to document the support of the modeled conformations with omit electron density maps (Supplementary Figures 2 and 3). As we noted above and in the manuscript, the electron density quality over the tail regions fluctuates significantly, but there is information content (supported by the omit maps), which importantly illustrates the character of the H1

tails. In any case, however, we want to ensure a clear perspective for the reader. As such, we have added a further discussion of the tails following the second paragraph of the Discussion section: “Indeed, the LH tails display conformational and electron density characteristics closely analogous to that of the core histone tails²⁴, which is consistent with a general activity of the histone tails in electrostatically-driven stabilization of nucleosome binding and chromatin compaction. The richness in secondary structure-disrupting, proline/glycine, residues appear to foster the conformational character of the H1x tails, which are largely composed of random coil structural elements in the H1x-169a crystals. While the two copies of the H1_{XNT/CT} structures are overall highly distinct, the H1_{XCT} displays molten globule-like conformations that are consistent with the LH_{CT} being an intrinsically disordered domain that undergoes a disorder-to-order transition or condensation upon binding nucleosomes (characterized for H1.0)³². In any case, there are apparently alternative tail structures sampled in the H1x-169a crystals and most likely multiple, if not many, distinct conformations that would be available to the tails in other nucleosomal contexts (*i.e.*, in chromatin).”

27. 5. Regarding the binding data shown in Fig. S16:

(i) *Why does panel (d) only show the molar input ratios 5.25-6 for H1x and not 1-6 as in panel (c) ?*

(ii) *The standards shown on the right-hand side of panel (c) cover only the low end of the range of band intensities observed on the left-hand part of the gel. Was the standard curve used in Table S4 generated from only this limited set of data ? (Notably, the condensate binding curve in Figure 7a shows a conspicuous jump between $rH1 = 3.5$ and 4 for H1.0. Might this reflect a quantitation error based on an extrapolated standard curve?)*

Response: In order to allow for complete and redundant coverage of all the samples, we had to run many gels, from which standard slopes were calculated and applied to the condensate samples. Given the number of data points required to explore a finely sampled saturation curve, it was not possible to obtain a full range of samples and standards within a single gel. We therefore analyzed many gels and fit condensate samples to the most proximal standard slopes. To establish further confidence in our findings, we went carefully back through all of the condensate experiments and were able to locate additional gels that we have now included in the full analysis. While this has not resulted in any significant changes in the final values, it further confirms the accuracy of the experimental approach. Nonetheless, we have accordingly updated panel a of Figure 8 and Supplementary Tables 4-7. In addition, we are providing the collection of gels used in the analysis as Supplementary Data 2. Notably, the most conspicuous feature, that of the H1x ‘jump’ between 3.5 to 4.0, is still apparent. It is even more pronounced when looking only at data points within a single gel (15951; Supplementary Table 5), whereby the two 3.5 data points are 1.2 and 1.5 (average=1.4), while that of the 4 data point is 3.2. This represents a slightly greater jump (1.8) than for the whole data set average (1.5).

28. 6. Many of the structural figures are somewhat difficult to interpret. In certain cases, it was necessary to view the atomic coordinates of the deposited PDB file in order to better understand the figure. A small inset showing a global view of the dinucleosome might help orient the reader.

Response: We have revised Figure 5 to include a panel (5a) to orient the viewer with respect to the H1 binding context that is detailed. In addition, an animation to support Figure 2b (Supplementary Movie 1) has been included that should provide further orientational value overall as well as for Figure 5 (also referred to in the Fig. 5 legend).

RESPONSES TO REVIEWER COMMENTS (Manuscript NCOMMS-23-39237A)

We owe substantial gratitude to the reviewers, who went to great lengths to evaluate our work and recommend improvements. The actions undertaken and the resulting current manuscript address all the concerns and incorporate all the suggested revisions of both reviewers. Moreover, the manuscript has been reconfigured from scratch to include additional biophysical experiments and an exhaustive re-analysis, yielding a streamlined presentation with decisive conclusions. The structural issues that were outlined by reviewer #2 are all rectified, and the data associated with the five unpublished X-ray crystal structures, which are part of the submission, are freely accessible at the Protein Data Bank platform (PDB codes and structural summary now provided in Table 1). Below we outline the revisions implemented for each of the points raised by the reviewers.

In response to comments by Reviewer #1:

1. The authors have adequately addressed all of our queries and comments in the revised MS. However, I would suggest the authors include a table of all 10 constructs that yielded well-diffracting crystals to complement the initial part of the Results, providing general information about each. This would help readers keep straight the PDB identifiers for each, which contained H1s, what H1 they contained, which H1s had sufficient densities for modeling and which did not, what H1 binding modes were detected in each structure, and overall resolutions for each of the 10. It would be helpful to also list the 14 constructs that did not yield well-diffracting crystals, perhaps in the table legend.

Table 1 has been added to the manuscript. It summarizes all (10) of the linker histone-nucleosome assembly crystal structures (encompassing 6 different DNA constructs), where one or more linker histone molecules were explicitly modeled. The other constructs, which are described in reference 21, are referred to in the table legend.

2. I will also note that H1s forming general bridging interactions between multiple DNA segments was first elucidated by Jean Thomas and David Clark (cf JMB 1986 187:569; EJB 1988 178:225). Given the multiple DNA bridging modes by H1s described in the current MS, it seems appropriate to cite this work.

These two works have been incorporated into the discussion and cited as references 36 and 37.

In response to comments by Reviewer #2:

Synopsis:

Moving forward with this manuscript necessitates the following steps.

- i) The authors should correct their H1.0-169an structure (pdb 7xvl) by repositioning helix a3 in the same register as that of Zhou et al (pdb 7k5x).*
- ii) They should correct their H1x-169a structure (pdb 7xx7) by removing the NTD and CTD, and by removing all core histone tail residues that are dubious according to current best crystallographic practice.*
- iii) They should obtain new PDB validation reports for the updated structures and provide these together with the updated coordinates upon manuscript resubmission.*

iv) They should revise their manuscript accordingly by removing or modifying any text, tables or figures that refer to the H1 helix $\alpha 3$, to the H1 NTD and CTD, or to the core histone tails. This includes deleting most if not all of the "Tails Provide Electrostatic Glue" section.

3. i) *The authors should correct their H1.0-169an structure (pdb 7xvl) by repositioning helix $\alpha 3$ in the same register as that of Zhou et al (pdb 7k5x).*

The H1.0-169an structure (7XVL), as well as the H1.0-169a structure (6LAB), have been amended and re-refined so that the $\alpha 3$ helix register is identical to that of the published H1.0/H5 ‘consensus’ structures and all our other H1.0 and H5 structures.

4. ii) *They should correct their H1x-169a structure (pdb 7xx7) by removing the NTD and CTD, and by removing all core histone tail residues that are dubious according to current best crystallographic practice.*

The H1x-169a structure (formerly 7XX7) was amended and re-refined, and the H1x-169a X-ray data was extended to include all the recorded low-resolution terms (2.7–92 Å; the resolution range was originally 2.7–20 Å). A total of 587 protein residues were removed from the model, including nearly all of the core histone and linker histone N- and C-terminal tail regions. Only short motifs of the H1x NTD (34-44, chains U and V) and CTD (117-120, chain U; 117-127, chain V), as they emerge from the globular domain (GD, residues 45-116), remain in the final model. All of the H2A C-terminal tails have also been removed from the model, and 7 of the 8 core histone N-terminal tails have been removed from the model. The final model and revised X-ray data set were deposited at the *PDB* and released under the new entry code, 8YTI.

5. iii) *They should obtain new PDB validation reports for the updated structures and provide these together with the updated coordinates upon manuscript resubmission.*

The three revised structures (8YTI, 7XVL, and 6LAB) as well as all the other structures from this work are freely accessible on the *PDB* platform. The accessible information includes validation reports in addition to a variety of available data and assessment tools associated with each of the depositions.

6. iv) *They should revise their manuscript accordingly by removing or modifying any text, tables or figures that refer to the H1 helix $\alpha 3$, to the H1 NTD and CTD, or to the core histone tails. This includes deleting most if not all of the "Tails Provide Electrostatic Glue" section.*

All of the data and the structures have been analyzed de novo, and the manuscript was rewritten accordingly. All of the former references that were made to the structural elements, which are now removed from or revised in the models, are absent in the current manuscript.

RESPONSES TO REVIEWER COMMENTS (Manuscript NCOMMS-23-39237B-Z)

We thank the reviewers for their valuable time and input on our manuscript.

In response to comments by Reviewer #2:

1. *However, the second category of concerns remains unresolved, as the manuscript still does not establish that the structural observations made in this highly engineered system have meaningful implications for how linker histones function in vivo.*

Importantly, however, these findings do not demonstrate that the linking modes observed in the crystal structures are used, or are even accessible, in physiological chromatin. The data and arguments presented in support of biological relevance are not sufficiently persuasive. In particular, the observation that condensates can bind linker histones at stoichiometries above 1 is of uncertain biological significance. (The authors point to the existence of cell types with H1:nucleosome ratios above 1 as supporting evidence, but the cited reference [Woodcock et al., 2006] reports H1:nucleosome ratios below 1 in most systems, with values slightly above 1 observed only in highly specialized contexts). More importantly, the array experiments do not show that H1 within the condensates adopts the L1 or L2 binding modes reported in the crystal structures, raising the possibility that these modes may be in vitro phenomena specific to their crystallization system.

In conclusion, while the structural analyses have some interest, the study does not appreciably advance our understanding of linker histone function in vivo, which limits its overall impact for the chromatin field.

We agree that these are crucial considerations. We should note, however, that a linker histone:nucleosome stoichiometry greater than unity is not, a priori, a prerequisite for the in vivo relevance of the linking modes. In fact, a recent cryo-EM analysis of tetra-nucleosome arrays, having different and physiologically relevant nucleosome repeat lengths (NRLs), showed that certain NRLs are refractory to H1 binding, even under saturating conditions (H1:nucleosome molar stoichiometry of 1.2; Dombrowski et al., 2022). This suggests that a subset of nucleosomes in the cell do not allow on/off-dyad H1 binding, which would effectively increase the pool of available H1. In any case, we mean to argue that a >1 bound H1:nucleosome stoichiometry would necessitate additional linker histone-nucleosome interaction types (i.e., linking modes) beyond the conventional on/off-dyad modes. The quantification of isolated cellular chromatin studies, which have shown that certain cell types display >1 chromatin-bound linker histone stoichiometry (Bates & Thomas, 1981; Woodcock et al., 2006), would mean that the alternative linker histone binding modes would have to be relevant within at least certain cellular contexts. Our array condensate data, showing that all the somatic linker histone variants are capable of high stoichiometry nucleosome binding in vitro, lend support to this H1 activity and demonstrate potential structural (chromatin compacting/condensing) consequences that moreover support the X-ray structures, and vice versa. Since we are concerned that we may not have made this main argument based on the >1 stoichiometry clear, and in conjunction with the related comments by Reviewer #3, we have added a clarification paragraph to the Discussion.

Regarding the main aspect as to the biological relevance of the linking modes, and more specifically whether they may coincide with crystalline-state artifacts, it is a reasonable concern that we also shared, at least initially. However, as we have tried to emphasize in the manuscript, the nucleosomal crystals are analogous to the crowded molecular environment and ionic (especially divalent metal cation) conditions within the nucleus. By analogy, one may consider the first structure of a nucleosome, a crystal structure of the linker DNA-lacking core particle

(NCP; Luger et al., 1997, and further details with Davey et al., 2002), which displayed association of the H4 tail with the H2A/H2B-acidic patch and non-specific DNA binding of the disordered histone tails, both presenting as crystal (internucleosomal) contacts. Moreover, DNA twist defects (aka, DNA stretching) were also observed as a phenomenon presumably stabilized, if not induced, by crystal contacts. However, all three of these crystal structure observations have ultimately been established as biologically relevant. In our engineered systems here, most importantly, the crystalline context allows for the possibility to both support and observe multi-nucleosome binding by an individual linker histone molecule. In contrast, cryo-EM structures of linker histone-nucleosome assemblies are obtained by using very ionic strength and divalent metal-free conditions in order to prevent nucleosome aggregation on the grid, which correspondingly facilitates the single-particle reconstructions. In this manner, it is not possible to observe the linking modes since the conditions select against this. Along the same lines, short (3-, 4-, 6-) nucleosome arrays or 30 nm fiber structures in isolation would not be amenable to linking mode linker histone binding.

In terms of crystallographic approaches, only two nucleosome-linker histone structures have been reported outside of our work (Zhou et al., 2015 and Bednar et al., 2017). We infer that the scarcity of crystal structures for this key system is a reflection of the prevalence of linker histone binding diversity, which tends to thwart crystallization efforts. This was initially our motivation for engineering nucleosomal constructs that would promote lattice order by replacing the loosely stacking blunt-ended DNA termini with annealing single-stranded overhangs. In this sense, we believe that examples of the linking mode had not been reported earlier as an in vitro system was lacking, which was capable of both supporting and revealing this type of linker histone association.

In response to comments by Reviewer #3:

2. *Do the various linker histone binding modes observed have any biological significance? The large majority of the literature supports a stoichiometry of up to one H1 molecule per nucleosome. Consequently, chromatin structures with more than one H1 per nucleosome, such as those presented in this paper, might not be biologically important. However, there is one clear and relevant example of chromatin with more than one H1 molecule per nucleosome: transcriptionally inert chicken erythrocyte chromatin has ~0.9 molecules of H5 and ~0.4 molecules of H1 (Bates & Thomas, 1981; PMID 7312631). In that paper, the stoichiometry of linker histone to core histones was also measured very carefully for lymphocyte and glial chromatin, finding one H1 per nucleosome in both cases. This paper could be cited as support for the biological relevance of models with more than one linker histone per nucleosome (at least for those including H5). In their Discussion, the authors note that even if the average stoichiometry is one linker histone per nucleosome, H1 molecules might not be distributed evenly in chromatin, resulting in some regions with two linker histones per nucleosome and others with none. This suggestion is supported by refs. 34 and 35.*

Thank you for pointing out the underlying reference, which provides more detail to these important observations. The Bates & Thomas, 1981 study has been added to the discussion as well as a further elaboration to ensure clarity in the arguments surrounding stoichiometry and linking mode association.

3. *The use of 'N' for 'nucleosome' does not seem necessary.*

“N” has been replaced with “nucleosome” throughout the text. We have left it in place in figure labelling (and references thereto) where it is necessary because of space limitations.

In my initial review of the manuscript by Adhireksan et al., I raised serious concerns about two of the authors' claims regarding their crystal structures. The first claim related to structural variability in the globular domain of linker histone H1.0; namely, that in one of the authors' structures (H1.0-160an, pdb 7xvl), a specific helix in this domain (helix $\alpha 3$) shows a one-residue shift in register relative to other structures described in the manuscript, as well as to previously published structures and to the model predicted by AlphaFold. The second disputed claim, pertaining to another of their structures (H1.x-169a, pdb 7xx7), was that the authors could successfully model the N-terminal domain (NTD) and C-terminal domain (CTD) of histone H1x as structurally well-ordered domains.

The first claim is extraordinary because, without a substantial change in stereochemical environment or physicochemical conditions, there is no reason to expect a helix within a stably folded domain to undergo a single-residue shift in packing register, whereas incorrectly assigning the helical register is a common error in the interpretation of medium-resolution maps. The second claim is even more extraordinary since the approximately 40 and 90 residues of the NTD and CTD, respectively, are known to be intrinsically disordered. Remarkably, the authors affirm that the two copies of each of these domains in their crystal structure fold into well-ordered conformations that are unique for each copy, devoid of secondary structure, and buttressed by relatively few contacts with DNA or with other (non-tail) structured histone residues. Whereas concerns about the first claim have only minor implications for the overall conclusions of the study, concerns about the second are profoundly significant because they necessitate a major revision of the manuscript.

In my initial evaluation, I examined electron density maps generated from diffraction data and atomic coordinates provided by the authors. Based on this analysis, I found no evidence to support either of the claims mentioned above. Surprisingly, in their response letter, the authors reaffirmed both claims and dismissed my concerns as a matter of differing opinions. However, my concerns are based on objective criteria that are widely accepted in macromolecular crystallography field. For the benefit of the authors, these are explicitly detailed below for each of the claims.

Claim 1: Shift of helical register.

1. The model is inconsistent with omit map density. From the data supplied by the authors (pdb 7xvl), I calculated omit maps after deleting helix $\alpha 3$ from the model. One such map is displayed below. The side chain density for helix $\alpha 3$ is weak and ambiguous, and hence cannot be relied upon to confidently assign the correct register. Notably, however, the map distinctly reveals that the backbone atoms of residue Thr78, located at the bottom of the helix, lie completely outside the density (red arrow). In contrast, the structure previously reported by Zhou et al. (pdb 7k5x), in which the corresponding helix is shifted upwards by one residue, agrees well with the density in this region. The omit map shown below was generated using the program Phenix, but similar results were obtained using program Refmac. Thus, the authors' configuration of helix $\alpha 3$ is clearly inconsistent with omit map density.

Fo-Fc Omit map (at 3.0 sigma and 1.5 sigma)

Stereo view of omit map calculated from atomic coordinates and structure factors supplied by the authors (pdb 7xvl) using program Phenix version 1.20.1 after deleting residues spanning H1.0 helix α 3 (chain o, residues 57-81) and performing coordinate and B-factor refinement to remove model bias.

2. The corrected model refines better. I generated a modified model of the authors' structure by replacing their coordinates for helix α 3 by the corresponding coordinates of Zhou et al (pdb 7k5x), followed by coordinate and B-factor refinement of the two models in Phenix. As seen from the plot below, the correlation coefficient between the resulting electron density and residues in helix α 3 was considerably higher for the modified model (mean CC=0.869) than for the original model (mean CC=0.826), confirming that the authors' configuration for this helix is incorrect.

3. The corrected model is stereochemically more plausible. In the authors' model, the side chains of residues Gln67 and Arg74 on helix α 3 are situated within a relatively hydrophobic environment, in which their respective OE1 and NH1 side chain atoms (marked by red stars in the figure below) could only fulfill their hydrogen bonding requirements by forming a C-H bond with an aliphatic carbon atom. While not impossible, this is considerably less favorable than forming an O-H or an N-H bond. In contrast, the modified model places a hydrophobic residue (Ile68 or Leu75) in the corresponding

position, while orienting Gln67 and Arg74 towards the solvent and adjacent to the DNA phosphate backbone, which is clearly a more favorable environment.

4. The authors appear to have made the same error previously. In their response letter, the authors note that one of their previous structures, reported in Adhireksan et al., 2021, displays the same register for helix $\alpha 3$ as the disputed structure (7xvl) from the current manuscript. The prior structure (pdb 6lab) includes two independent copies of histone H1.0. An omit map calculated from the PDB data deposited by the authors reveals that, in one of the H1.0 molecules (chain V), residue Thr78 lies entirely outside the density (red arrow), mirroring the finding for their present 7xvl structure. The overall density is much weaker for the other H1.0 molecule (chain U), in which helix $\alpha 3$ was modeled in a different conformation from that in chain V. However, this conformation is implausible because helix backbone carbonyl groups point directly at each other and because Thr78 is a Ramachandran outlier. Thus, the configuration of helix $\alpha 3$ appears incorrect in the previous 6lab structure and is presumably the source of error in the current 7xvl structure.

Fo-Fc Omit map for pdb 6lab (at 3.0 sigma and 1.5 sigma)

Claim 2: The NTD and CTD adopt well-ordered conformations in the H1x-169a structure.

1. Omit map density does not justify modeling of the NTD and CTD. The authors provide stereoviews of omit maps as evidence to support their modeling of the H1x NTD and CTD. However, given the relatively featureless appearance of the density, the low threshold at which the Fo-Fc maps are contoured (1.0 sigma, a level at which considerable noise may occur), the limited views shown, and the low-complexity sequence being traced, these figures are inadequate for allowing a critical evaluation of this claim. I therefore calculated omit maps from the data provided by the authors using both programs Phenix and Refmac. After deleting the NTD (res. 1-45) and CTD (res. 117-213), coordinate and B-factor refinement was performed to remove any model bias. In the resulting omit maps, one observes clear density for a few residues immediately preceding and following the globular domain (res. 41-45 and res. 117-120) in the Fo-FC maps contoured at 1.5 and 2.0 sigma. However, there is no convincing density for any of the other residues in these domains, at any contour level. Examples are shown below for chain U, and similar density is observed for chain V. Thus, contrary to the authors' assertions, these independently calculated omit maps do not support the model.

2. NTD and CTD residues correlate poorly with the (non-omit) electron density.

The Real-space R-value Z-score (RSRZ) measures how well residues fit into the electron density at a given resolution. A residue is considered an RSRZ outlier if its RSRZ value is above 2, and is highly suspect if its RSRZ is above 3. Whereas the PDB validation report supplied by the authors shows that most H1 residues in the globular domain have good RSRZ values, nearly all residues in the NTD and CTD are RSRZ outliers (red dots in the figure below). Moreover, a large number of residues (≥ 17 in the NTD and ≥ 30 in the CTD) have an RSRZ between 5 and 14.6, indicating a very poor fit to the map.

[REDACTED]

3. A model lacking the NTD and CTD agrees better with the diffraction data. I generated a modified version of the authors' structure by deleting the NTD and CTD and refined the original structure and the modified version. Using either Phenix or Refmac, the modified model exhibits a substantially better R_{free} value both before and after refinement, with a decrease ranging between 0.41% and 0.63%. Thus, inclusion of the NTD and CTD results in a poorer agreement with the crystallographic data.

	R_{work} (init)	R_{free} (init)	R_{work} (final)	R_{free} (final)
PHENIX				
a. Original	0.2332	0.2682	0.2129	0.2700
b. Δ NTD/ Δ CTD	0.2328	0.2629	0.2139	0.2659
Difference (b - a)	-0.0004	-0.0053	0.0010	-0.0041
Refmac				
c. Original	0.2317	0.2705	0.2135	0.2694
d. Δ NTD/ Δ CTD	0.2308	0.2647	0.2126	0.2631
Difference (d - c)	-0.0009	-0.0058	-0.0009	-0.0063

4. The geometry of the NTD and CTD is exceptionally poor. The PDB validation report for 7xx7 supplied by the authors indicates severe issues with the geometry of these domains. Specifically, 3 NTD residues (all in chain U) and 13 CTD residues (6 in chain U, 7 in chain V) are outliers in the Ramachandran plot. A total of 18 NTD residues (7+11 in chains U+V) and 61 CTD residues (29+32 in chains U+V) show unusually close contacts, with a clash overlap ranging from 0.5 Å to 1.28 Å. Furthermore, 35 NTD residues (20+15 in chains U+V) and 53 CTD residues (31+22 in chains U+V) exhibit non-rotameric side chains, placing this histone structure at the 0th percentile compared to all other X-ray entries in the PDB and to all others at a similar resolution, i.e., placing it among the poorest structures ever deposited.

5. The two copies of each domain share no local conformational similarity. The authors note that the two copies (chains U and V) of the NTD or CTD exhibit different overall conformations. This is not inherently unusual, since intrinsically disordered domains are known to adopt distinct conformations in different structural contexts. However, in such cases one typically observes the conservation of one or more short motifs across structures. What is surprising about the authors' model is the total absence of such structural motifs: there is no similarity in the local conformation of residues between the two copies. This is evident in the Kleywegt plots, which show very different (ϕ , ψ) backbone angles for corresponding residues in the two copies, and in the local RMSD plot, which compares the two structures over a sliding window of 9 residues. The figure underneath shows local alignments for a few short stretches from chains U and V, including those with the lowest RMSD, highlighting the significant differences in conformation.

6. The analogy with previous structures is unpersuasive. In their response letter, the authors state that "in the H1x-169a structure, the linker histone tails display structural and electron density characteristics that are fully analogous to that seen previously with the core histone tails at high and lower resolutions (i.e., $\sim 2\text{--}3\text{ \AA}$; for example, Davey et al., 2002), whereby the tails undergo multiple disorder-to-order transitions as they 'meander' through the lattice."

This argument is relatively weak. Consider, for example, the H3 tail from the NCP147 structure (chain E in PDB 1kx5) from the 2002 study cited by the authors. While the absence of diffraction data from the PDB entry precludes an examination of electron density, one can nevertheless assess the stereochemical plausibility of the modeled H3 tail conformation. In this structure, the H3 residues pass through a narrow solvent channel in the crystal lattice and are extensively buttressed by symmetry related molecules. Nearly all these tail residues are within contact distance of well-ordered DNA or protein residues (see Figure), and so the tail conformation is highly plausible. This scenario contrasts sharply with the present case, where the H1x NTD and CTD residues have considerably fewer spatial constraints, allowing them great freedom to adopt multiple conformations. For instance, only about a third of residues in the CTD of chain U are within contact distance (4.5 \AA) of DNA or of any other structured protein residue (excluding other NTD, CTD or core histone tail residues) in the crystal lattice. Additionally, there are extensive stretches of residues with no such contacts at all (e.g., res. 129-145, 177-187, 193-209 in chain U; res. 118-141 and 179-188 in chain V). Moreover, many backbone atoms in the CTD do not fulfill their hydrogen bonding potential. In essence, the CTD adopts a compact, spaghetti-like coil conformation but it is unclear what maintains most of these residues in position.

7. The authors' modeling of core histone tails is also unjustified. Omit maps calculated after deleting the core histone tails from pdb 7xx7 (approximately res. 1-12, 1-27, 1-34, 1-19 for the various copies of H2A, H2B, H3 and H4, respectively) reveal that the density does not support the modeling of any of these tails. An example is provided below for the H3 tail (chain A). The PDB validation report shows that most of these residues exhibit poor RSRZ values. Refinement performed with Refmac demonstrates a marked improvement (0.8%) in the R_{free} value upon removal of the core histone tails from the model. Hence, the authors appear to have overfitted their electron density maps not only for the linker histone but also for the core histone tails.

	R_{work} (init)	R_{free} (init)	R_{work} (final)	R_{free} (final)
Refmac				
a. Original	0.2317	0.2705	0.2135	0.2694
b. $\Delta\text{NTD}/\Delta\text{CTD}$	0.2308	0.2647	0.2126	0.2631
c. $\text{NTD}/\Delta\text{CTD}/\Delta\text{all tails}$	0.2297	0.2570	0.2111	0.2550
Difference (c-b)	-0.0011	-0.0077	-0.0015	-0.0081
Difference (c-a)	-0.0020	-0.0135	-0.0015	-0.0144

pdb 7xx7
chain A

2.5 sigma

1.5 sigma

2.0 sigma

1.0 sigma

Conclusion. The authors' assertions regarding the variability in the register of H1 helix $\alpha 3$ and their ability to model the H1 NTD and CTD are contradicted by the evidence and arguments provided above. Best crystallographic practice was clearly not adhered to: the map density was over-interpreted, the model was refined with exceptionally poor geometry, and warnings from PDB validation reports were ignored. The notion that the H1x-169a crystal structure has captured an intrinsically disordered domain as large as the H1 CTD in two distinct, well-ordered and compact conformations challenges conventional understanding of such domains. Surprisingly, the authors seem unaware of the extraordinary nature of this claim and of the need for compelling evidence to substantiate it.

Moving forward with this manuscript necessitates the following steps.

i) The authors should correct their H1.0-169an structure (pdb 7xvl) by repositioning helix $\alpha 3$ in the same register as that of Zhou et al (pdb 7k5x).

ii) They should correct their H1x-169a structure (pdb 7xx7) by removing the NTD and CTD, and by removing all core histone tail residues that are dubious according to current best crystallographic practice.

iii) They should obtain new PDB validation reports for the updated structures and provide these together with the updated coordinates upon manuscript resubmission.

iv) They should revise their manuscript accordingly by removing or modifying any text, tables or figures that refer to the H1 helix $\alpha 3$, to the H1 NTD and CTD, or to the core histone tails. This includes deleting most if not all of the "Tails Provide Electrostatic Glue" section.

I have taken exceptional pains to substantiate my concerns with this manuscript. I trust that the authors will now give due consideration to these concerns and address them diligently.